# OBJECT-ORIENTED TRANSITION MODELING WITH INDUCTIVE LOGIC PROGRAMMING

## ABSTRACT

Building models of the world from observation, i.e., *induction*, is one of the major challenges in machine learning. In order to be useful, models need to maintain accuracy when used in novel situations, i.e., generalize. In addition, they should be easy to interpret and efficient to train. Prior work has investigated these concepts in the context of *object-oriented representations* inspired by human cognition. In this paper, we develop a novel learning algorithm that is substantially more powerful than these previous methods. Our thorough experiments, including ablation tests and comparison with neural baselines, demonstrate a significant improvement over the state-of-the-art. The source code for all of our algorithms and benchmarks will be available online after publication.

## 1 INTRODUCTION

Learning is a vital part of any intelligent agent's behavior. In particular, one of the most important problems in the field of machine learning is *induction*: constructing general rules from examples, allowing an agent to explain its observations and make predictions in the future. Induction is also an important part of how humans – and humanity as a whole – acquire knowledge. While much of science takes the form of induction, it also encompasses other learning activities, such as categorizing shapes, playing a new video game, and even practicing physical skills. To enable these feats, the human implementation of induction has several essential traits. First, we create models that *generalize*, i.e., make accurate predictions in new situations. Second, our models are *interpretable*, so they can be reasoned about and communicated to others. Third, we learn *efficiently*, in terms of both number of observations and computational power.

To make induction tractable, humans think of the world in terms of objects and their relationships, which allows for efficient learning and generalization of knowledge to new situations (Spelke, 1990). In artificial intelligence, *object-oriented* representations seek to capture this insight by representing an agent's perception of the world as a set of objects, each consisting of a type and a collection of numerical vector attributes (Diuk et al., 2008; Stella & Loguinov, 2024). The object-based representation serves as a middle-ground between low-level (sensory) input, for which learning structured rules remains intractable (Locatello et al., 2020), and high-level (relational) formulations, which use a large amount of domain knowledge to simplify the task structure (Garrett et al., 2020). This makes object-oriented inductive learning an important, but challenging, problem.

One particularly important form of induction is learning about the dynamics of a system, i.e., discovering physical laws from observation. This can be modeled using the Markov Decision Process (MDP) framework (Sutton & Barto, 2018). In this setting, the agent interacts with an unknown environment by taking actions $a$ in response to observations of the environment's state $s$. Each action causes the environment to transition to a new state $s'$ according to its *transition function $T$*, such that $s' = T(s, a)$. In learning the dynamics of an MDP, the agent's objective is to create a model $\hat{T}$ that produces the same outputs as the environment's ground-truth $T$. This learning process occurs *online*, meaning that the agent must refine its model continuously as it receives a stream of observations. In addition, the agent should be able to learn the dynamics of various environments *without* extensive domain-specific tuning.

Traditionally, learning a transition model might mean enumerating the values of $T$ Sutton et al. (1999). However, by using a structured state format – e.g., as a set of objects with attributes – we

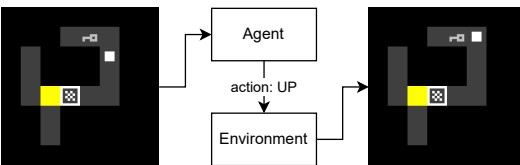

| object | type | pos | status | locked |
|--------|--------|--------|--------|--------|
| 16 | wall | (4, 2) | - | - |
| 50 | player | (5, 2) | - | - |
| 51 | door | (3, 4) | - | true |
| 52 | key | (4, 1) | free | - |
| 53 | goal | (2, 4) | - | - |

(a) an example transition in the `keys` domain  (b) partial object listing for the initial state in (a)

Figure 1: An example game transition with a partial listing of the initial state in the object-oriented representation. Although we use meaningful names to increase readability, the information given to an agent does not contain these labels.

can instead represent the MDP *intrinsically* by implementing $T$ as a program operating on states in this representation. This leads to the possibility – and accompanying challenges – of *generalization*, as the use of an implicit representation allows for a potentially infinite space of states and transitions to be defined by a single program of finite size. The agent's model of the world should allow it to accurately predict the outcome of *all of these possible transitions*.

Because very little prior work has studied this problem, most existing induction methods have significant limitations. For example, the most popular approach in modern machine learning is *deep learning*, which uses artificial neural networks to approximate arbitrary functions (Schrittwieser et al., 2020). However, as shown by prior work and our own experiments, they do not learn efficiently or generalize reliably (Nagarajan et al., 2021; Zhang et al., 2023; Mirzadeh et al., 2024). In addition, although their parameters could be considered "learned rules" in the induction sense, they are difficult to interpret (Ghorbani et al., 2019; Druce et al., 2021).

To overcome these limitations, Stella & Loguinov (2024) introduced QORA, which uses an Inductive Logic Programming (ILP)-based framework to tackle the problem of object-oriented transition learning. This yields interpretable models, but as we discover through experiments in several novel domains, their ILP method does not converge in many cases. Unfortunately, other existing ILP algorithms are not suitable replacements, as the application to object-oriented prediction imposes several requirements. First, the majority of prior ILP methods only support batch-mode learning (Cropper & Dumančić, 2022); using these algorithms in the online setting would require rebuilding each rule from scratch every time a new observation is made, which is intractable. Second, whereas we seek algorithms that do not require manual input of domain knowledge, many existing approaches need extensive domain-specific configuration in the form of, e.g., meta-rules (Cropper & Tourret, 2020) or architecture selection (Dong et al., 2019). Although the TG algorithm meets these two requirements (Driessens et al., 2001), it conducts continuous-valued scalar regression, which makes it inapplicable to object attribute vector prediction.

In this paper, we introduce *TreeLearn*, a novel ILP algorithm that is well-suited to use with the object-oriented transition learning framework. Our method conducts statistically-guided induction of logical programs, incrementally building more-complex models as necessary to achieve better prediction accuracy. TreeLearn models take the form of *first-order logical decision trees* (FOLDTs) (Blockeel & De Raedt, 1998), a highly expressive and interpretable representation for inductive models, allowing us to tackle a variety of complex domains. Using TreeLearn, we build *TreeThink*, an object-oriented transition learning algorithm that efficiently and reliably produces models that generalize strongly to novel transitions within their environment. We demonstrate the efficacy of our approach with a thorough empirical evaluation, including ablation tests and comparison with sophisticated neural baselines.

## 2 TREETHINK

In this section, we describe our algorithm, *TreeThink*, which provides two high-level interfaces: `observe`, which is used to train the learner, and `predict`, which queries the model. The observation function takes transition triples consisting of a state, an action, and the resulting next state $(s, a, s')$ such that $s' = T(s, a)$. The prediction function uses the learned model to compute $\hat{T}(s, a)$. Both functions operate on object-based states, such as those shown in Figure 1.

Figure 1a shows an example of a transition in one of our domains, called `keys`. In this environment, which we use as a running example throughout this section, the agent controls a player character moving through a maze-like area containing keys, doors, and goals. The doors, initially locked, can be opened by bringing an unused key to them. Any key can open any door, but each key can only be used once. The agent receives a penalty for each move, with a larger penalty for attempting to make an illegal move (e.g., bumping into a wall or locked door). A reward is given if the player character ends up on a goal.

Figure 1b shows a subset of the transition's initial state in the object-oriented representation, using human-readable labels for clarity (compare to Figure 6 in Appendix C). The state $s$ consists of a set of some number $n_s$ of objects. Each object belongs to a class, e.g., `player` or `wall`, and has some attributes, e.g., `position` (shortened to `pos`) and `color`. We use $s.c$ to refer to the subset of objects in $s$ that have class type $c$. Each of an object's attributes has some value, which is a vector of integers, e.g., $(5, 2)$; the length of the vector is determined by the attribute it corresponds to. We use the notation $s_i$ to refer to object $i$ in state $s$ and $s_i[m]$ to refer to the value of attribute $m$ of that object. The notation $X[m]$ is also used when an object is labeled $X = s_i$. We denote by $class(s, i)$ and $attr(s, i)$ the class of object $i$ and its set of attributes, respectively. Any reward signal $R$ is folded into the state transition function $T$ through inclusion of a special object of class `game` with a single attribute called `score`, which tracks the cumulative sum of rewards. Thus, no separate reward model is necessary.

With this framework, we can formulate the model as an algorithm that predicts the change in each object's attribute values from $s$ to $s'$. TreeThink breaks this down using a collection of subroutines, which are called *rules* (Stella & Loguinov, 2024). Each rule $\hat{T}_{c,m,a}$ predicts the changes for a particular attribute $m$ in objects of a specific class $c$ when a certain action $a$ is taken. For example, one rule may predict the player's position when the `right` action is taken, while another could predict the `game` object's `score` attribute when the `up` action is taken. This allows us to simplify the model while retaining generality, as individual rules are each typically small, but any particular rule can still be highly complex if necessary. The process of prediction using rules is shown in Figure 2b. We represent these rules using *First-Order Logical Decision Trees* (FOLDTs).

## 2.1 FIRST-ORDER LOGICAL DECISION TREES

FOLDTs are an extension of classical decision trees to the setting of first-order logic. They are highly expressive, allowing us to model rules for complex domains, and interpretable, making it easy to decode the knowledge they learn through training (Blockeel & De Raedt, 1998). A FOLDT representing a rule from the `keys` domain is shown in Fig. 2a. The tree takes as input a state and a "target" object to make predictions for. It then uses information from the state to produce an output for the target object, i.e., to determine how one of its attributes should change. The top box (labeled 0) shows metadata about this tree's rule: its input object $X_0$ is a `player` and it predicts how that player's position changes when the `RIGHT` action is taken. Evaluation proceeds recursively, starting from the root.

Each branch of the tree (numbered $> 0$) contains a *test*, which evaluates to either true or false based on conditions in the current state. These tests are logical formulas that refer to properties of, or relations between, objects. The tests are existentially quantified, meaning that they pass if *any* objects exist that satisfy the condition. If a test passes, the left branch is taken and the quantified variable(s) are bound. If the test fails, any variable appearing in a quantifier in that test is not bound (since no such object exists). This means that, e.g., the $X_1$ in box 2 ($\exists X_1 \in$ doors) is the same as in box 3 ($X_1[\text{open}] = 1$), but not the same as the one in box 1 ($\exists X_1 \in$ walls). In tests without quantifiers, it is implicit that *any* satisfying binding is acceptable; e.g., if the test in box 1 fails, then as long as there is *any* door next to the player (box 2) that also is open (box 3), the latter will test pass and the tree will return $(1, 0)$. Thus, a test only fails if there is no possible binding that will satisfy it. When a leaf node is reached, the value (or distribution of values) in that leaf is returned. The procedure encoded by this FOLDT is equivalent to the program shown in Figure 2c.

The tree we have shown outputs constants at its leaves, which do not depend on the player's current position. To use these values for our rules, we treat them as *deltas*. Thus, if $F_{c,m,a}$ is the FOLDT for a rule $\hat{T}_{c,m,a}$, then

$$\hat{T}_{c,m,a}(s, s_i) = s_i[m] + F_{c,m,a}(s, s_i). \tag{1}$$

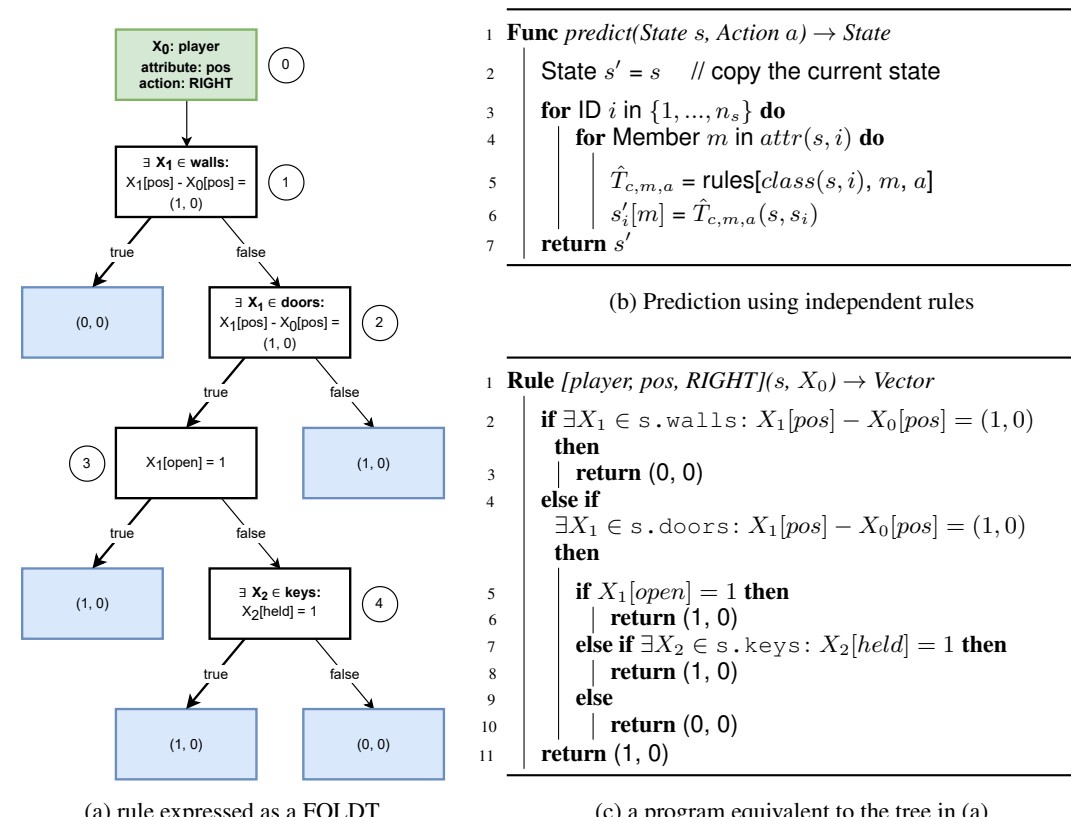

(a) rule expressed as a FOLDT

```
1  Func predict(State s, Action a) → State
2      State s′ = s    // copy the current state
3      for ID i in {1, ..., n_s} do
4          for Member m in attr(s, i) do
5              T̂_{c,m,a} = rules[class(s, i), m, a]
6              s′_i[m] = T̂_{c,m,a}(s, s_i)
7      return s′
```

(b) Prediction using independent rules

```
1  Rule [player, pos, RIGHT](s, X_0) → Vector
2      if ∃X_1 ∈ s.walls: X_1[pos] − X_0[pos] = (1, 0)
       then
3          return (0, 0)
4      else if
       ∃X_1 ∈ s.doors: X_1[pos] − X_0[pos] = (1, 0)
       then
5          if X_1[open] = 1 then
6              return (1, 0)
7          else if ∃X_2 ∈ s.keys: X_2[held] = 1 then
8              return (1, 0)
9          else
10             return (0, 0)
11     return (1, 0)
```

(c) a program equivalent to the tree in (a)

Figure 2: FOLDT and equivalent program representation of a rule for the `keys` domain.

The last factor to settle is the kind of conditions, which we also call *facts*, that can be used as tests. We use the same two types as QORA: *attribute equality*, of the form

$$P_{m,v}(i)\colon X_i[m] = v, \tag{2}$$

and *relative difference*, of the form

$$P_{m,v}(i, j)\colon X_j[m] − X_i[m] = v. \tag{3}$$

The space of tests can be augmented with more varieties, but these two domain-agnostic condition classes are sufficient for all those that we have conducted experiments in. Even the $(c, m, a)$ rule structure could be implemented, in part, as tests in the tree; however, enforcing rule separation in the way we do improves both efficiency and interpretability.

## 2.2 TreeLearn

We now describe how to construct FOLDTs from examples. The approach we take is *top-down induction*, similar to classical decision trees Quinlan (1986), which grows the tree (starting from a single leaf node) by splitting leaves based on some information metric. Our incremental algorithm does this through a recursive process with three steps, beginning at the tree's root each time an observation is received. First, statistics for all candidate tests being tracked at the current node are updated. Second, the test used for the node may be updated. If the current node is a leaf, we check if it should be split into a branch; if it is already a branch, we check if it should use a different test. Third, the algorithm recurses to the appropriate child based on the evaluation of the branch's test. We next cover each of these steps in more detail.

**Updating Tests**  Our algorithm keeps track of candidate tests and corresponding statistics in each node. A separate candidate is created for each observed fact and every arrangement of variables

(both existing, from higher levels of the tree, and new) as arguments to the condition. The information kept for each candidate consists of a table of counters, incremented for each observed sample, indexed in one dimension by whether the test was true or false and in the other by the output for that sample. This can be treated as a joint probability distribution, over which an information metric can be computed to assess the utility of the test.

**Updating Nodes**    The goal of the algorithm is to eventually converge to a stable tree structure that tests only the information that is necessary to determine the outcome of each transition. To accomplish this, we allow existing branches to change their test over time. However, this requires resetting the tree nodes below that branch, which slows learning. Thus, we need a test evaluation method that allows us to ensure that leaves and branches are only modified when there is a high level of certainty that the new test will improve the model's performance. For this, we use the predictive power score introduced by Stella & Loguinov (2024).

For a test with joint probability distribution $\hat{P}$, which distinguishes conditions in a set $X$ and predicts outputs from the observed set $Y$, the score $\mathcal{S}$ is

$$\mathcal{S} = \sum_{(x,y)\in X\times Y} \hat{P}(y|x)\hat{P}(x,y), \tag{4}$$

which gives the test's expected confidence in the correct output on a randomly-sampled input. This score takes values in $[0, 1]$, where 1 means the predictions are perfectly accurate. To evaluate tests, we compute a confidence interval over their $\mathcal{S}$ scores. The interval sizes are controlled by a single confidence-level hyperparameter, $\alpha$. When a test's confidence interval is greater than (not overlapping) the current test, it becomes the node's new test. For a leaf node, the initial test is an uninformed baseline. Leaves become branches when any test surpasses the baseline.

This confidence-based testing is also key to TreeLearn's ability to model stochastic transition functions. Algorithms that learn decision trees for deterministic processes typically continue refining the model until each leaf contains only a single class. However, we are also interested in modeling *stochastic* domains, in which the same condition may lead to more than one outcome. In this case, no test will reliably give better predictions than the baseline, so our learning process will stop while one or more leaves still contain multiple output values. Instead of yielding the most-common value (or sampling from the observed values), our model returns the entire distribution of whichever leaf it reaches. This enables us to faithfully reconstruct the probability distribution of stochastic transition functions *and* to interpret the model's uncertainty during learning in deterministic environments.

**Recursion**    The last step of the algorithm is to recurse down the tree, passing the observation sequentially to every node in the path determined by each branch's current test. To ensure correctness when updating tests in each node and its descendants – i.e., so that each test can be evaluated correctly – we compute the set of all satisfying bindings at every branch that is visited. When a left-branch is taken, this set is modified to include new variables and ensure each binding satisfies the branch's test.

## 2.3 INFERENCE OPTIMIZATIONS

Evaluating a FOLDT for inference (i.e., prediction) proceeds recursively, similar to the learning process. However, when the tree is not being trained, there are two major optimizations that can be used to drastically improve the efficiency of tree evaluation. The first is *on-demand queries*. Instead of computing every true fact from the object-oriented state as input to the ILP model, these can be checked only as-needed (and then cached) when evaluating branch tests. Since most facts are not used during inference, this leads to substantial savings, as only a small portion of the objects need to be processed. The second is *short-circuit branch evaluation*. Rather than explicitly computing the set of all satisfying bindings, the tree can be evaluated in a depth-first style. Specifically, any time a binding is found that allows a left-branch to be taken, the algorithm can immediately recurse; if that binding allows the node's descendants to also take left-branches, then evaluation can immediately return, passing the appropriate output value back to the current node's parent. This both reduces memory overhead and enables the algorithm to terminate as soon as a binding is found that leads to a "preferred" path through the tree.

We have included complete pseudocode for our algorithm in Appendices A (ILP) and B (object-oriented interfaces). We next discuss the results of our empirical evaluations.

## 3 EXPERIMENTS

We conduct experiments in three groups: first, comparison against the prior state-of-the-art in object-oriented transition modeling, QORA (Stella & Loguinov, 2024); second, ablation and performance tests, investigating details of TreeThink's operation; third, evaluation of a sophisticated neural-network baseline. In many experiments, we also include a "naive" baseline that we call `static`, which is a model implementing $\hat{T}(s, a) = s$ (i.e., it predicts that nothing ever changes). We use the Earth Mover's Distance (EMD) state-distance metric described by Stella & Loguinov (2024) to evaluate model error, for which a value of zero indicates perfect accuracy. To ensure good coverage, we run tests in a variety of domains.

### 3.1 ENVIRONMENTS

We conduct tests in nine domains and one "domain set", which is a parameterized meta-domain used to study the scaling properties of a learning algorithm. Several of these environments come from (or are based on domains from) Stella & Loguinov (2024). Domains that have no reward signal are marked "not scored". In our evaluation, we also evaluate on altered versions of some domains in which the reward signal has been erased (marked "-scoreless") and transfer to larger, more complex instances (marked "-t"). More details (and sample images) for all environments are given in Appendix C; here, we briefly describe domains for which we include experimental results in the main text. `fish` is a stochastic environment in which the agent must estimate the conditional distribution of fish movements (not scored). `maze` is an extension of the `walls` domain (Stella & Loguinov, 2024) that adds goal objects and a reward signal that encourages short paths. `coins` is a Traveling Salesman-style routing problem that takes place inside of a maze full of coins to be picked up. `keys` is a maze task in which the goal may be blocked behind one or more locked doors, which can be opened by picking up keys. `switches` is a combination of the `walls` and `lights` environments (Stella & Loguinov, 2024), where the player must navigate through a maze to toggle lights remotely (not scored). $scale(n_p, n_c)$ is a combination of the `moves` and `players` domain sets Stella & Loguinov (2024), which augments the `walls` domain with $n_p$ independent player objects, each of which has $n_c$ copies of each movement action (not scored).

### 3.2 COMPARISON WITH PRIOR WORK

Consider Figs. 3a-3f, showing TreeThink and QORA learning in several domains. In all of these environments, TreeThink rapidly converges to zero error. On the other hand, QORA never achieves perfect accuracy in any of the domains with reward signals. Notably, because the reward signal is transparently folded into the transition function, this limitation of QORA is not unique to reward modeling; any particularly complex rules – such as those in `keys-scoreless` and `switches` – appear to be impossible for QORA to learn. Certain rules are also significantly more challenging for QORA to learn, such as those in `coins-scoreless`, where it takes approximately four times longer than TreeThink to converge.

The root cause for these issues, which TreeLearn addresses, seems to be twofold. First, QORA's ILP method is unable to express formulas with nested quantifiers, which are necessary to represent rules such as the one shown in Figure 2c. In contrast, FOLDTs can nest quantifiers arbitrarily deep. Second, QORA has no variable binding process, which leads to difficulty resolving rules with two or more conditions involving the same quantified object, such as with boxes 2 and 3 in Figure 2a. In contrast, our algorithm keeps track of variable bindings as part of each hypothesis, allowing it to differentiate between a new test using a *previously-bound* object and a new test using a *newly-bound* object. This enables TreeLearn to determine the utility of new conditions more quickly and reliably.

Moving on to Figs. 3g-3j, we find that TreeThink's learning process is highly stable, while QORA suffers from significant performance issues when faced with complex rules. In several of our experiments (including the one shown in Figs. 3i and 3j), QORA's resource usage suddenly increases, leading to a crash. In environments such as `fish`, the $\alpha$ hyperparameter had to be reduced to prevent this behavior, while ThreeThink reliably converges even with a relatively high value of $\alpha = 0.01$.

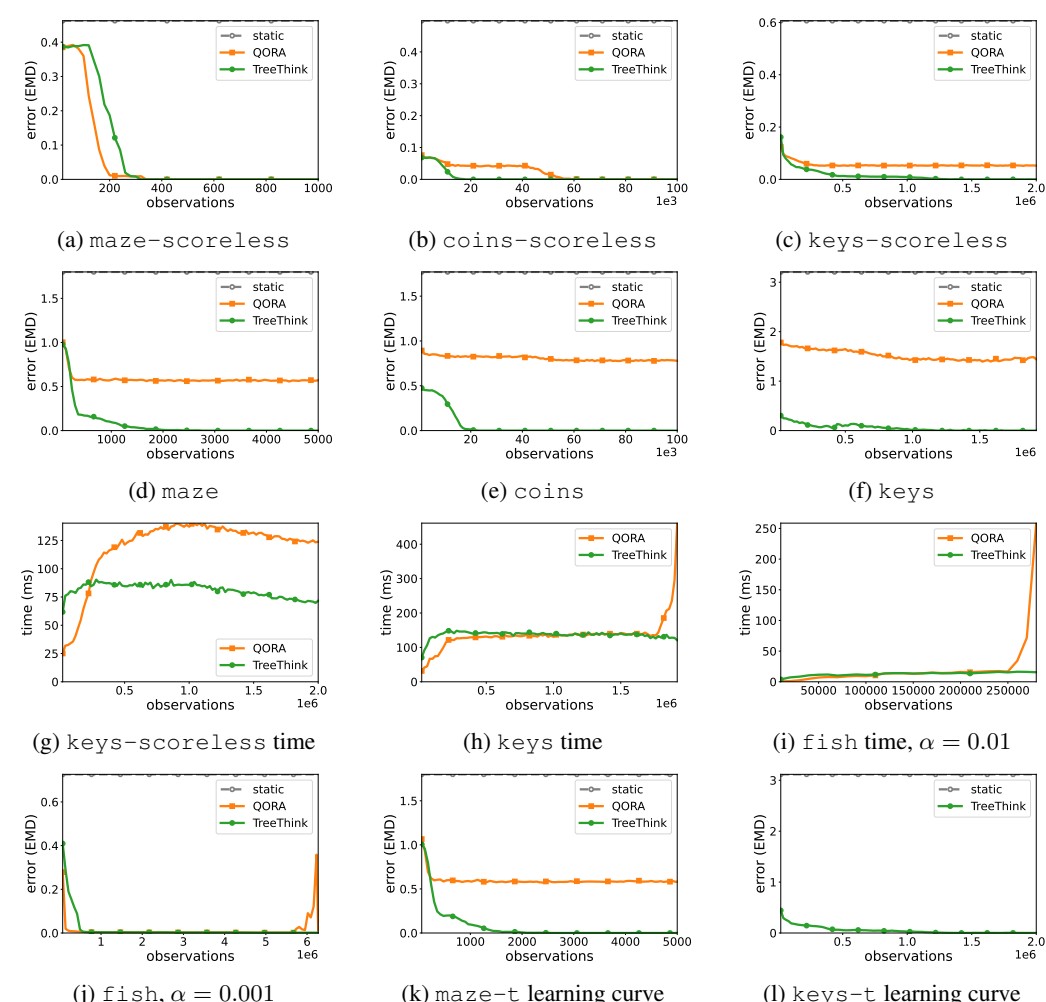

Figure 3: TreeThink vs. QORA predictive modeling

Lastly, we conduct transfer learning experiments to larger levels in the `maze`, `coins`, and `keys` domains. Specifically, training occurs in $8 \times 8$ grids, while evaluation is done in $32 \times 32$ grids. Results for `maze` and `keys` are shown in Figures 3k and 3l, respectively. As expected, TreeThink displays perfect generalization, as its learned models align with the ground-truth transition dynamics. This can be verified easily by inspecting the models; examples of FOLDTs learned by TreeThink are shown in Appendix D, Fig. 19. For many other additional results, see Appendix D.

## 3.3 ABLATION AND PERFORMANCE TESTS

We next conduct ablation tests to analyze the impact of our inference optimizations and branch updating process. We then demonstrate some interesting properties of TreeThink's performance.

**Inference Optimizations**  We test four settings of the two inference optimizations: `none` (no optimizations to inference), `eval` (short-circuit tree evaluation), `query` (on-demand state queries), and `both` (optimizing state queries and tree evaluation). Figure 4a shows that these optimizations do not negatively impact the learning process, as expected; when given the same data, the training proceeds identically regardless of the inference optimization setting. Figure 4b demonstrates the massive performance boost to inference (i.e., the `predict` function) that is given by the optimizations. In small levels ($8 \times 8$) in the `maze` domain, inference with both optimizations is approx. $34\times$ faster than without either. We conducted additional experiments in `switches` and `keys` with

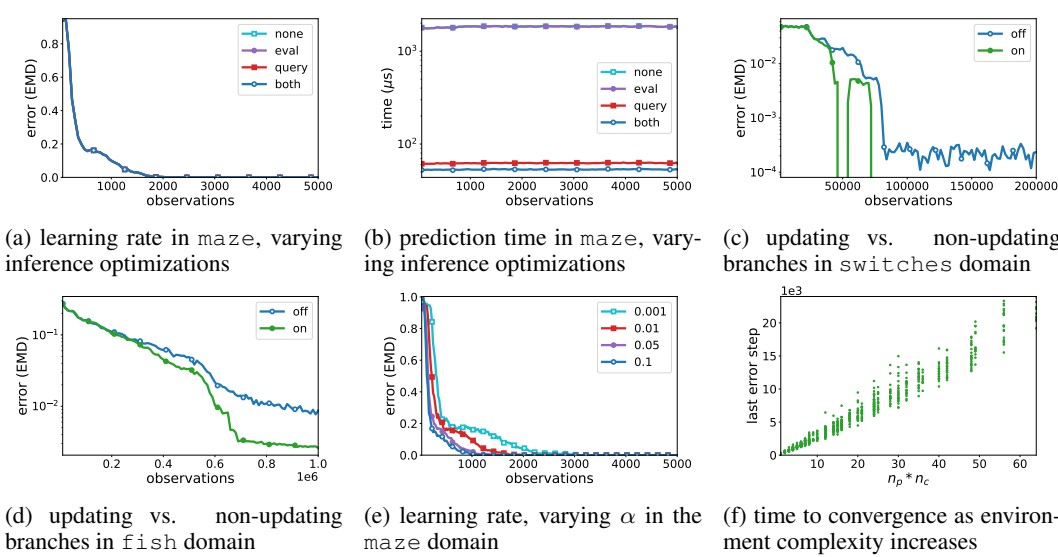

(a) learning rate in `maze`, varying inference optimizations

(b) prediction time in `maze`, varying inference optimizations

(c) updating vs. non-updating branches in `switches` domain

(d) updating vs. non-updating branches in `fish` domain

(e) learning rate, varying $\alpha$ in the `maze` domain

(f) time to convergence as environment complexity increases

Figure 4: Ablation and Performance Tests

similar results. Most importantly, the performance boost increases with the number of objects in the state; in `maze-t`, we observe a speedup of $880\times$ compared to running with no optimizations. The results of the tests in these domains are shown in Appendix E.

**Branch Updating** The purpose of branch updating is to ensure that, as the agent continues to receive data, each level of the tree has the opportunity to make use of the best possible test. If a branch's test is fixed upon creation, then it is possible for spurious correlations to lead to incorrect tree formation, i.e., incorporating unnecessary information. Thus, updating branches should improve the algorithm's convergence, both in rate and stability – which is what we find, as shown in Figures 4c and 4d, where breaks in a curve indicate zeroes.

**Hyperparameter Robustness** When learning algorithms have hyperparameters, it can be difficult to apply them to new problems. Neural networks, for example, have many hyperparameters; in addition, their performance is often highly sensitive to the specific values of these hyperparameters (Adkins et al., 2024). TreeThink, on the other hand, has only a single hyperparameter, $\alpha \in (0, 1)$. Fortunately, as shown in Figure 4e, our algorithm operates well across a wide range of values. In all of our other experiments throughout this paper, we use $\alpha = 0.01$ (unless otherwise specified), which leads to rapid and stable convergence in both the simpler domains and the highly complex ones.

**Performance Scaling** One of the core motivations behind using program induction for object-oriented transition learning is that it should allow the agent to scale much more efficiently. While our previous experiments showed that TreeThink scales to larger levels, it is also interesting to note how the learning rate (i.e., sample complexity) scales with the complexity of the environment's transition function. For example, if an environment has many actions that behave almost identically, the agent's learning rate should scale linearly with the number of actions (as it can learn each independently). This is exactly what we find in experiments in the `scale`$(n_p, n_c)$ domain set, where we can vary $n_p$ and $n_c$ to arbitrarily increase the environment's complexity without qualitatively changing the dynamics. Figure 4f shows the results of our experiment, in which we track the last observation on which TreeThink makes an error in prediction (i.e., the number of steps before it fully converges) for many runs using randomly sampled $n_p$ and $n_c$, each in $\{1, ..., 8\}$.

### 3.4 NEURAL BASELINES

Neural networks have become a popular tool for nearly every induction task, including ones involving object-oriented representations. For example, Chang et al. (2016) introduced the *Neural Physics*

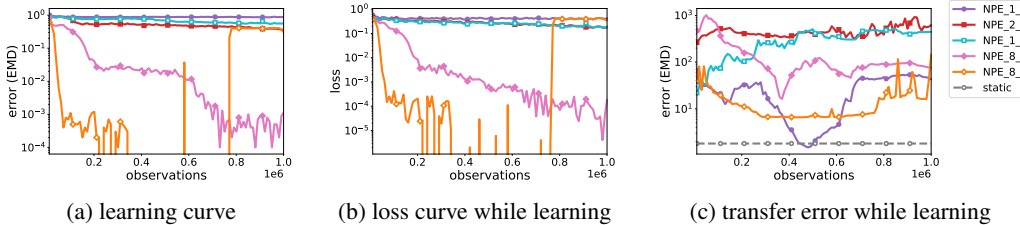

(a) learning curve          (b) loss curve while learning          (c) transfer error while learning

Figure 5: Using various NPE architectures to model the `maze` domain. Semilog-$y$ plots are used to better visualize small values. Breaks in a line (e.g., the error of NPE_8_3) indicate zeroes.

*Engine* (NPE) for modeling physical dynamics. Stella & Loguinov (2024) evaluated an NPE architecture for object-oriented transition learning, finding that it fails to reach zero error in the `walls` domain. However, it is unclear whether their network design had sufficient capacity to represent the environment's transition function *in principle*. Thus, inspired by Zhang et al. (2023), we create a custom NPE architecture with hand-tuned weights that achieves perfect accuracy (zero error) on every possible transition in the `maze` domain. We then train randomly-initialized copies of this network, as well as larger variations of it, to investigate whether the training process can discover weights that generalize. More details and results are included in Appendix F.

To match the tests in the prior subsections, we train the networks in $8 \times 8$ levels. We denote by NPE_X_Y a network $X$ times wider than our hand-crafted design with $Y - 1$ extra layers in each final feed-forward block. Shown in Fig. 5a, we find that NPE_8_3 is seemingly able to achieve perfect accuracy after approx. $340K$ observations, which is about $200\times$ slower than TreeThink in this domain. As displayed in Fig. 5b, this coincides with the network reaching zero training loss (to measured precision). However, it eventually (after about 770K observations) encounters a state in which it produces a significant error, after which its performance immediately rises and remains at the same level as the other networks. We suspect that there are at least two causes: first, the optima found by the training process performs well in many cases, but poorly in others, though this fact may not be apparent even after thorough testing; and second, the gradient of the loss is very steep near these optima, so small deviations can lead to parameter updates that significantly diminish the network's accuracy on almost all transitions.

During training, we also test each network in $16 \times 16$ instances of the same environment. While the networks manage to easily surpass the `static` baseline in the $8 \times 8$ levels, Fig. 5c shows that they almost always output predictions with *massive* error in these new levels; only NPE_1_1, which gets the highest training error, dips below the `static` line for a brief moment. In other words, the networks' knowledge does not transfer – even to levels only slightly more complex. This also significantly impacts the agent's ability to plan successfully; see Appendix G for experiments comparing TreeThink and NPE using Monte-Carlo Tree Search (Schrittwieser et al., 2020).

# 4 FUTURE WORK

We introduced *TreeThink*, a new object-oriented transition learning algorithm capable of modeling more-complex environments than prior work, including domains with reward signals. TreeThink is based on our novel ILP algorithm, *TreeLearn*. To facilitate reproduction and extension of our results, the code for our algorithms, neural baselines, and benchmarks will be released after publication.

Our contributions open up several paths for future work. First, it will be worthwhile to investigate potential runtime optimizations, especially to the `observe` function. Second, extension to even more kinds of environments should follow naturally from the framework we have outlined. Third, a theoretical analysis of the convergence and sample complexity of TreeThink would be extremely worthwhile. Fourth, as TreeThink represents the first object-oriented transition learning algorithm capable of modeling reward functions, our work enables future developments in planning for object-oriented reinforcement learning.

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

## A FOLDT LEARNER PSEUDOCODE

To improve portability and usability, our implementation of TreeThink's FOLDT-learning ILP method is self-contained. For completeness, we describe all of the components of our codebase here. We use Python-esque syntax for types; e.g., `MyClass[T]` refers to a generic (i.e., templated) class with a single type parameter, `T`.

### A.1 UTILITY CLASSES

We make use of several general-purpose utility classes and functions. As the implementations are typically straightforward, we generally include only the interface, though pseudocode implementations are given for more complex components.

#### A.1.1 TREE CLASS

We use a templated binary tree class as the basis for our first-order logical decision trees. The class has two template parameters: the datatype stored in each branch, which we denote by B, and the datatype stored in each leaf, which we denote by L. As the implementation is fairly straightforward and typical, we simply enumerate the interfaces in Algorithm 1. For conciseness, we refer to the type `Tree[B, L]` as `Tree`, as the template parameters remain the same throughout.

---

**Algorithm 1:** The interface to a generic binary tree class

1 **Func** *Tree.init(B data, Tree left, Tree right) → Tree*
   | // Initialize a branch with two children

2 **Func** *Tree.init(L data) → Tree*
   | // Initialize a leaf

3 **Func** *Tree.isBranch() → bool*
   | // Test if this tree node is a branch

4 **Func** *Tree.getBranchData() → B*
   | // If this tree is a branch, get its branch data

5 **Func** *Tree.getLeftChild() → Tree*
   | // If this tree is a branch, get its left child sub-tree

6 **Func** *Tree.getRightChild() → Tree*
   | // If this tree is a branch, get its right child sub-tree

7 **Func** *Tree.isLeaf() → bool*
   | // Test if this tree node is a leaf

8 **Func** *Tree.getLeafData() → L*
   | // If this tree is a leaf, get its leaf data

9 **Func** *Tree.convertToLeaf(L data) → void*
   | // Turn this tree node into a leaf with the specified data; any existing child sub-trees are
   |   deleted

10 **Func** *Tree.convertToBranch(B data, Tree left, Tree right) → void*
   | // Turn this tree node into a branch with the specified data and children

---

#### A.1.2 JOINT PROBABILITY DISTRIBUTION

We use a class template called `FTable`, shown in Algorithm 2, to manage the joint probability distribution associated with each test (as well as the baseline inside each leaf). The class is a template parameterized by the input space $X$ (representing the possible values of the test, e.g., true and false) and output space $Y$ (representing the set of outputs that have been observed). Note that the implementation of this class determines the input and output space sizes dynamically, as observations are received.

---

**Algorithm 2:** The interface to our conditional probability distribution class, FTable

---

1  **Func** *FTable.init() → FTable*
   | // Initialize an empty joint probability distribution

2  **Func** *FTable.observe(X input, Y output) → void*
   | // Record a single (input, output) observation $(x, y) \in X \times Y$

3  **Func** *FTable.getConditionalDistribution(X input) → ProbabilityDistribution[Y]*
   | // Return the conditional distribution $\hat{P}(y|x)$ for a specified input value $x \in X$

4  **Func** *FTable.getScoreinterval() → ConfidenceInterval*
   | // Compute a confidence interval over this joint probability distribution's $\mathcal{S}$ score (Equation 4)

---

### A.1.3  CONFIDENCE INTERVAL

We use a class with data members `lower` and `upper` to represent confidence intervals. The only additional noteworthy component is the non-overlapping comparison we use, as shown in Algorithm 3.

---

**Algorithm 3:** The interface for a simple confidence interval structure

---

1  **Func** *ConfidenceInterval.init(float lower, float upper) → ConfidenceInterval*
   | // Initialize a confidence interval with $0 \leq$ lower $\leq$ upper $\leq 1$

2  **Func** *isBetterThan(ConfidenceInterval a, ConfidenceInterval b) → bool*
   | // Is interval `a` greater than (not overlapping) interval `b`?
3  | **return** a.lower $>$ b.upper

---

### A.1.4  PREDICATES

Several classes are shown in Algorithm 4. The `Predicate` class represents a type of predicate, e.g., $P(i)\colon X_i[pos] = (1, 0)$. The GroundPredicate structure represents a predicate with objects bound to its arguments. The predicate set classes are used to input truth values to FOLDTs.

---

**Algorithm 4:** Utility classes related to predicates

---

// An abstract class, extended by specific predicate types (e.g., attribute equality, relative difference)

1 **class** {

2     **Func** *getArgumentCount() → int*
    | // The arity of this predicate

3     **Func** *getArgumentTypes() → list[int]*
    | // The class type of each argument to this predicate

4 } Predicate

// A predicate, along with object bindings to each of its argument slots

5 **struct** {

    // The (lifted, with no bound variables) predicate type of this ground predicate

6     Predicate $p$

    // The ids of the objects that are bound as input to the predicate $p$

7     list[int] arguments

8 } GroundPredicate

// Stores the truth value of (ground) predicates, computed from an object-based state, for use by FOLDTs

9 **class** {

10     **Func** *getValue(GroundPredicate g) → bool*
    | // Check whether the given fact (predicate evaluated on specific objects) is true

11     **Func** *getObservations(Predicate p) → set[GroundPredicate]*
    | // Get all of the ground predicates for a predicate type $p$
    | // This corresponds to all object bindings that make the predicate true in the current state

12 } AbstractPredicateSet

// Explicitly lists all true predicates from an object-based state

13 **class** {

14     **Func** *add(GroundPredicate g) → void*
    | // Store the fact that $g$ is true

15     **Func** *getPredicates() → set[Predicate]*
    | // Enumerate all the types of predicates that have true bindings (used to optimize FOLDT observation)

16 } FullPredicateSet extends AbstractPredicateSet

// Implements the on-demand predicate query optimization: only predicates that are used by a FOLDT get computed
// When the predicate set is queried by a FOLDT, it checks its cache;
// If the cache is missing an entry, the predicate set will evaluate the truth-value directly from the object-based state.

17 **class** {

18     **Func** *init(State s) → QueryPredicateSet*
    | // Initialize the predicate set with an empty cache

19 } QueryPredicateSet extends AbstractPredicateSet

---

### A.1.5 COMBINATORIAL FUNCTIONS

We use several combinatorics functions, e.g., to generate combinations and cartesian products. The most important functions for tree building are `newvars` and `bindings`, which are used to enumerate all of the ways that existing (and new) variables can be bound into a predicate of a given arity. The functions are shown in Algorithm 5.

---

**Algorithm 5:** The combinatorics functions we use

---

1 **Func** *Combinatorics.combinations(int n, int r) $\rightarrow$ list[list[int]]*
  | // Compute all choices of $r$ elements from the set $\{0, ..., n - 1\}$

2 **Func** *Combinatorics.product(int n, int r) $\rightarrow$ list[list[int]]*
  | // Compute the Cartesian product $\{0, ..., n - 1\}^r$

3 **Func** *Combinatorics.product(list[set[int]] sets) $\rightarrow$ list[list[int]]*
  | // Compute the Cartesian product of the sets in a list
  | // If the input is a list of sets $[X_1, X_2, ..., X_n]$, then the output contains all elements in the set
  |   $X_1 \times X_2 \times ... \times X_n$

4 **Func** *Combinatorics.newvars(int k) $\rightarrow$ set[tuple[list[int], int]]*
  | // Computes all ways to generate new variables to fill in $k$ argument slots (up to the
  |   equivalence of new variables)
  | // For example, $k = 1$ gives: $(X_0)$; $k = 2$ gives: $(X_0, X_0)$, $(X_0, X_1)$;
  | // and $k = 3$ gives: $(X_0, X_0, X_0)$, $(X_0, X_0, X_1)$, $(X_0, X_1, X_0)$, $(X_0, X_1, X_1)$, $(X_0, X_1, X_2)$
  | // The function returns a set of tuples, each containing a list of variable ids and the number of
  |   new variables in that list
5 | set[tuple[list[int], int]] listings = newvars($k - 1$)
6 | set[tuple[list[int], int]] result = {}    // Initialize empty set
7 | **for** (listing, $n$) in listings **do**
  |   | // Add all combinations up to $n$
8 |   | **for** $i$ in $\{0, ..., n - 1\}$ **do**
9 |   |   | list[int] vars = copy(listing)    // new list with same contents
10 |  |   | vars.append($i$)
11 |  |   | result.insert((vars, $n$))
  |   |
  |   | // Add listing with a new variable
12 |  | listing.append($n$)
13 |  | result.insert((listing, $n + 1$))
14 | **return** result

15 **Func** *Combinatorics.bindings(int e, int n) $\rightarrow$ set[tuple[list[int], int]]*
  | // Computes all ways to generate bindings for a predicate with $n$ arguments, using up to $e$
  |   existing variables
  | // The function returns a set of tuples, each containing a list of variable ids and the number of
  |   new variables in that list
16 | set[tuple[list[int], int]] result = {}    // Initialize empty set
17 | **for** $m$ in $\{0, ..., n\}$ **do**
18 |   | list[list[int]] slots_new = combinations($n, m$)    // Each list[int] specifies which slots will get
  |   |     a new variable
19 |   | list[list[int]] existing_vars = product($e, n - m$)    // Which existing variables will we use?
20 |   | set[tuple[list[int], int]] new_vars = newvars($m$)    // How will we bind new variables?
21 |   | **for** $s_{new}$ in slots_new **do**
22 |   |   | $s_{exist} = \{0, ..., n - 1\} \setminus s_{new}$
23 |   |   | **for** ($v_{new}, k$) in new_vars **do**
24 |   |   |   | **for** $v_{exist}$ in existing_vars **do**
25 |   |   |   |   | list[int] args = [0, 0, ...]    // List initialized to length $n$
  |   |   |   |   |
  |   |   |   |   | // Fill in the slots with both existing and new variables
26 |   |   |   |   | **for** ($i, v$) in zip($s_{exist}, v_{exist}$) **do**
27 |   |   |   |   |   | args[$i$] = $v$
28 |   |   |   |   | **for** ($i, v$) in zip($s_{new}, v_{new}$) **do**
29 |   |   |   |   |   | args[$i$] = $v + e$    // Offset new variable indices by the number of existing
  |   |   |   |   |   |     variables
30 |   |   |   |   | result.insert((args, $k$))
31 | **return** result

---

### A.1.6 FOLDT DATA CLASSES

The structures in Algorithm 6 hold data for the FOLDT learning and evaluation processes. Each branch node contains a Hypothesis (used as the test for that branch) and TrackingData (for continual updates); each leaf contains solely TrackingData (its baseline output distribution is used as the leaf's output).

---

**Algorithm 6:** The classes used to manage FOLDT data

---

1 **struct {**

    // The condition this hypothesis is testing
2     Predicate p

    // The ids of the variables that are input to this hypothesis' test's predicate condition
    // Note that this refers to variables from the tree's quantifiers, not to the ids of objects in a
      state
3     list[int] var_ids

    // The number of new variables this hypothesis' test introduces
    // All quantified variables have ids that count up starting from zero
4     int n_new_vars

    // The class type of each quantified variable *at* this node, using this test (inherits from parent
      nodes)
5     list[int] var_class_types

6 **} Hypothesis**

7 **struct {**

8     Hypothesis hypothesis

9     FTable counter

10 **} Candidate**

11 **struct {**

    // The number of variables that are *already* bound by the parents of this node
12     int n_existing_vars

    // The class type of each quantified variable *before* this node (inherits from parent nodes)
13     list[int] var_class_types

    // The set of predicates that have been observed and tracked, so they don't get
      double-tracked
14     set[Predicate] observed

    // List of hypotheses being evaluated, along with their observed joint probability distributions
15     list[Candidate] current

    // The baseline probability distribution of outputs observed at this node (not conditioned on
      any test)
16     FTable baseline

17 **} TrackingData**

18 **struct {**

    // The test used to make decisions at this branch
19     Hypothesis hypothesis

    // Tracking data, in case this branch needs to be updated
20     TrackingData tracking

21 **} Branch**

22 **struct {**

23     TrackingData tracking

24 **} Leaf**

---

## A.2 FOLDT CLASS

The FOLDT class, shown in Algorithm 7, ties all of the above pieces together to implement the observation and prediction interfaces. The FOLDT class is templated by its output type, $Y$. Functions related to observation are shown in Algorithms 8, 9, 10, 11, and 12. Functions related to prediction are shown in Algorithms 13 and 14.

---

**Algorithm 7:** The FOLDT class interface

---

1 **class {**

    // The learning hyperparameter, $\alpha \in (0, 1)$; we use a default value of $0.01$

2     float $\alpha = 0.01$

    // The number of objects this FOLDT takes as an arguent (for our purposes, this is always one)

3     int arg_count = 1

    // The class type of each input object for this FOLDT

4     list[int] arg_types

    // This FOLDT's internal binary tree, using our generic utility class

5     Tree[Branch, Leaf] tree

6     **Func** *init(float $\alpha$, int arg_count, list[int] arg_types) $\rightarrow$ FOLDT*
        | // Initializes a First-Order Logical Decision Tree

7     **Func** *reset() $\rightarrow$ void*
        | // Resets this FOLDT's internal tree back to a leaf with no recorded observations

8     **Func** *observe(PredicateSetFull observation, list[int] arguments, Y output) $\rightarrow$ void*
        | // Record a single observation

9         **return** observeRecursive(tree, observation, {arguments}, output)

10     **Func** *predict(AbstractPredicateSet observation, list[int] arguments) $\rightarrow$ ProbabilityDistribution[Y]*
        | // Compute a distribution over the output for a given input

11         result = evaluateShortCircuit(tree, observation, arguments)

12         **return** result[0].tracking.baseline.getConditionalDistribution(0)

13 **} FOLDT[Y]**

---

---

**Algorithm 8:** FOLDT class function: observeRecursive

---

1 **Func** *observeRecursive(Tree t, PredicateSetFull observation, set[list[int]] bindings_in, Y output) $\rightarrow$ void*

2     TrackingData tracking = ($t$.getBranchData().tracking **if** $t$.isBranch() **else** $t$.getLeafData().tracking)

    // Update current node

3     addPredicates(tracking, observation)

    // If this flag is true, we'll reset the node's children

4     bool new_test = updateTests(tracking, observation, bindings_in, output_value)

    // Convert leaf $\rightarrow$ branch or branch $\rightarrow$ leaf

5     **if** updateNodeType($t$, tracking) **then**

6         | new_test = false    // We don't want to reset the node's children twice

    // Reset or recurse

7     **if** $t$.isBranch() **then**

8         **if** new_test **then**

9             | resetBranch($t$, tracking)

10         **else**

11             (branch, bindings_out) = checkBranch(observation, $t$.getBranchData().hypothesis, bindings_in)

12             **if** branch **then**

13                 | observeRecursive($t$.getLeftChild(), observation, bindings_out, output)

14             **else**

15                 | observeRecursive($t$.getRightChild(), observation, bindings_out, output)

---

---

**Algorithm 9:** FOLDT class function: addPredicates

---

1 **Func** *addPredicates(TrackingData tracking, PredicateSetFull observation) → void*
2   **for** Predicate $p \in$ observation.getPredicates() **do**
3     **if** $p \notin$ tracking.observed **then**
4       tracking.observed.insert($p$)
      // Generate all candidates for this predicate (with all bindings of new and existing variables)
5       $n$ = $p$.getArgumentCount()
6       **for** (binding, $n_{new}$) $\in$ Combinatorics.bindings(tracking.n_existing_vars, $n$) **do**
        // Ensure this binding is consistent with the predicate's class restrictions
7         list[int] var_types = tracking.var_class_types
8         valid = true
9         **for** $i \in \{0, ..., n-1\}$ **do**
10           $v$ = binding[$i$]
11           class_restriction = $p$.getArgumentTypes()[$i$]
12           **if** $v <$ len(var_types) **then**
            // This variable may already have a type; maybe sure it's consistent
13             int var_type = var_types[$v$]
14             **if** var_type is None **then**
              // Not inconsistent yet, but may need to be restricted now
15               var_types[$v$] = class_restriction
16             **else if** class_restriction is not None **then**
              // Need to ensure that variable type and predicate restriction match
17               **if** var_type != class_restriction **then**
18                 valid = false
19           **else**
20             var_types.append(class_restriction)
21         **if** not valid **then**
22           **continue**
        // Add the new hypothesis
23         Hypothesis $h$
24         $h.p$ = $p$
25         $h$.var_ids = binding
26         $h$.n_new_vars = $n_{new}$
27         $h$.var_class_types = var_types
28         tracking.current.append(Candidate($h$, FTable()))

---

---

**Algorithm 10:** FOLDT class function: updateTests

---

1 **Func** *updateTests(TrackingData tracking, PredicateSetFull observation, set[list[int]] bindings_in, Y output) → bool*
2   tracking.baseline.observe(0, output)
3   **for** Candidate c $\in$ tracking.current **do**
4     branch = checkBranch(observation, c.hypothesis, bindings_in)
5     c.counter.observe(branch, output)    // `branch` will be either 0 (false) or 1 (true)
  // "Bubble up" the best hypothesis (sort, descending, by score intervals)
6   **for** $i \in$ [len(tracking.current) $- 2, .., 0$] **do**
7     Candidate a = tracking.current[$i$] Candidate b = tracking.current[$i+1$]
    // If `b` is better than `a` (with confidence), swap them (so `b` moves up in the list)
8     **if** isBetterThan(b.counter.getScoreInterval(), a.counter.getScoreInterval()) **then**
9       swap(a, b)
10       **if** $i = 0$ **then**
        // New best candidate; if this node is a branch, it will need to be reset
11         **return** true
12   **return** false

---

---

**Algorithm 11:** FOLDT class function: updateNodeType

---

1 **Func** *updateNodeType(Tree t, TrackingData tracking)* → *bool*
2    **if** $t$.isLeaf() **then**
       // Should we make this leaf back into a branch?
       // (i.e., is there a candidate with a confidence interval strictly greater than the baseline?)
3        **if** $\text{len}(\text{tracking.current}) > 0$ **and**
        tracking.current[0].counter.getScoreInterval() $>$ tracking.baseline.getScoreInterval() **then**
4           Candidate best = tracking.current[0]
5           $h$ = best.hypothesis
6           left = Leaf(TrackingData(tracking.n_existing_vars + $h$.n_new_vars, $h$.var_class_types, {},
           [], FTable()))
7           right = Leaf(TrackingData(tracking.n_existing_vars, $h$.var_class_types, {}, [], FTable()))
8           $t$.convertToBranch(Branch($h$, tracking), left, right)
9           **return** true
10    **else**
       // Should we make this branch back into a leaf?
11        **if** $\text{len}(\text{tracking.current}) = 0$ **or**
        tracking.current[0].counter.getScoreInterval() $\not>$ tracking.baseline.getScoreInterval() **then**
12           $t$.convertToLeaf(Leaf(tracking))
13           **return** true
14    **return** false // No update occurred

---

**Algorithm 12:** FOLDT class function: resetBranch

---

1 **Func** *resetBranch(Tree t, TrackingData tracking)* → *void*
2    Branch b = t.getBranchData()
3    Candidate best = tracking.current[0]    // Best test; we'll use it to reset the node and construct
       the children
4    $h$ = best.hypothesis

   // Update the branch to use the new best test
5    b.hypothesis = $h$

   // Reset the branch's children using the new test's bound variable information
6    $t$.left = Leaf(TrackingData(tracking.n_existing_vars + $h$.n_new_vars, $h$.var_class_types, {}, [],
     FTable()))
7    $t$.right = Leaf(TrackingData(tracking.n_existing_vars, $h$.var_class_types, {}, [], FTable()))

---

---

**Algorithm 13:** FOLDT class function: checkBranch

---

1 **Func** *matchVars(list[int] bindings, list[int] hvars, list[int] args) → (bool, list[int])*
 // Determine if additional variables can be matched to produce a consistent binding
2 **for** $i$ in $\{0, ..., \text{len}(args) - 1\}$ **do**
 // The index of the variable at this position in the hypothesis
 // E.g., if the hypothesis is $P(X_0, X_2)$ and $i = 1$, then $v = 2$
3 $v = hvars[i]$
 // The id of the object we are currently looking to bind
4 $o = args[i]$

 // Check if variable index is bound
5 **if** $v < \text{len}(bindings)$ **then**
 // If bound, object id must match
6 **if** $o \neq bindings[v]$ **then**
7 | **return** (false, [])
8 **else**
 // The variable isn't bound, so we can try binding it to this object

 // This only allows unique bindings (e.g., object B cannot be bound to both $X_0$ and $X_1$)
9 **if** $o \in bindings$ **then**
10 | **return** (false, [])
11 bindings.append($o$)
12 **return** (true, bindings)

13 **Func** *checkBranch(AbstractPredicateSet observation, Hypothesis h, set[list[int]] bindings_in) → (bool, set[list[int]])*
 // Determine whether left branch or right should be taken, based on available bindings and facts of the observation
14 set[list[int]] bindings_left  // Potential variable bindings, if the left branch can be taken
15 **for** list[int] bindings $\in$ bindings_in **do**
 // Looping over getObservations(predicate) automatically restricts the search to results with the right class types
16 **for** GroundPredicate $g \in$ observation.getObservations($h.p$) **do**
 // Check if this ground predicate's arguments are consistent with existing variable bindings
 // (and hypothesis variable indices)
17 (match, var_bindings) = matchVars(bindings, $h$.var_ids, $g$.arguments)
18 **if** match **then**
19 | bindings_left.insert(var_bindings)
20 **if** bindings_left is not empty **then**
 // The left branch can be taken, so we provide the new variable bindings
21 **return** (true, bindings_left)
22 **else**
 // If the right branch is taken, no new variables are bound
23 **return** (false, bindings_in)

---

---

**Algorithm 14:** FOLDT class function: evaluateShortCircuit

---

1 **Func** *evaluateShortCircuit(Tree t, AbstractPredicateSet observation, list[int] bindings) → (Leaf, bool, BitString)*

// An optimized version of `checkBranch` that evaluates recursively, in DFS order, and returns as soon as possible

2 **if** $t$.isLeaf() **then**
3    | **return** (t.getLeafData(), true, "1")

4 Hypothesis $h$ = t.getBranchData().hypothesis

5 (Leaf, bool, BitString) best_result = (None, false, "0")
6 best_rank = 0

7 **for** GroundPredicate $g \in$ observation.getObservations($h.p$) **do**
   // Check if this ground predicate's arguments are consistent with existing variable bindings
   // (and hypothesis variable indices)
8    (match, var_bindings) = matchVars(bindings, $h$.var_ids, $g$.arguments)    // See Algorithm 13
9    **if** match **then**
      // There is a match, so we can take the left branch
10       result = evaluateShortCircuit($t$.getLeftChild(), observation, var_bindings)
11       rank = result[2] + "1"    // Shift over the BitString by inserting a one at the end

12       **if** result[1] **then**
         // This child (and all of its children, recursively) got preferred paths; return immediately
13          **return** (result[0], true, rank)
14       **else**
         // Not preferred, but still good; see if it's better than the currently-most-preferred binding
15          **if** rank > best_rank **then**
16             best_rank = rank
17             best_result = result

// Can the left branch be taken?
18 **if** best_rank > 0 **then**
19    | **return** best_result

// No match, have to take right branch (not preferred)
20 result = evaluateShortCircuit($t$.getRightChild(), observation, bindings)
21 **return** (result[0], false, result[2] + "0")    // Shift over the BitString by inserting a zero at the end

---

## B  OORL LEARNER PSEUDOCODE

Algorithm 15 shows the TreeThink API, comprising two functions: `observe(s, a, s')` and `predict(s, a)`. Algorithm 16 shows the `extractFacts` function. Algorithm 17 shows how the `QueryPredicateSet` scans an object-based state to update its predicate cache. Recall that our implementation uses typed predicates (i.e., their argument slots are annotated with variable class types).

---

**Algorithm 15:** TreeThink's high-level `observe` and `predict` procedures

1 **Func** *observe(State s, Action a, State s') → void*

2    facts = extractFacts($s$)    // extract facts (ground predicates) from the object-based state

3    **for** $i$ in $\{1, ..., n_s\}$ **do**

4      **for** Member $m$ in $attr(s, i)$ **do**
      // calculate the change in attribute m's value, e.g., +(1, 0)

5        Value $v = s_i'[m] - s_i[m]$
      // learn to predict this attribute

6        Tree $t$ = rules[$class(s, i), m, a$]

7        $t$.observe(facts, $[i], v$)

8 **Func** *predict(State s, Action a) → State*

9    facts = QueryPredicateSet($s$)    // use on-demand predicate queries

10    State $s'$ = State()    // initialize empty state

11    **for** $i$ in $\{1, ..., n_s\}$ **do**

12      **for** Member $m$ in $attr(s, i)$ **do**

13        Tree $t$ = rules[$class(s, i), m, a$]
      // predict (a distribution over) this attribute's value

14        Value $v = t$.predict(facts, $[i]$)

15        $s_i'[m] = s_i[m] + v$

16    **return** $s'$

---

**Algorithm 16:** The `extractFacts` function

1 **Func** *extractFacts(State s) → FullPredicateSet*
  // For some class(es) of predicates, find all true bindings of those predicates in state $s$

2    FullPredicateSet facts    // initially empty

3    **for** $i$ in $\{1, ..., n_s\}$ **do**

4      $c_1 = class(s, i)$
    // Extract *attribute value* predicates

5      **for** Member $m$ in $attr(s, i)$ **do**

6        $v = s_i[m]$

7        $g$ = GroundPredicate($P_{m,v}(c_1\ X)\colon X[m] = v, [s_i]$)

8        facts.add($g$)

    // Extract *relative difference* predicates

9      **for** $j$ in $\{i + 1, ..., n_s\}$ **do**

10        $c_2 = class(s, j)$

11        **for** Member $m$ in $attr(s, i) \cap attr(s, j)$ **do**

12          $v = s_j[m] - s_i[m]$

13          $g$ = GroundPredicate($P_{m,v}(c_1\ X, c_2\ Y)\colon Y[m] - X[m] = v, [s_i, s_j]$)

14          facts.add($g$)

15    **return** facts

---

**Algorithm 17:** Updating the `QueryPredicateSet` cache; makes use of optimized state subsets (by class and by attribute value)

---

1 **Func** *scan(State s, Predicate p) → set[GroundPredicate]*

    // Find all of the variable bindings for which $p$ is true in state $s$

2     **if** $p$ is of type `attribute value`, $p = P_{m,v}(c_1\ X)$ **then**

3         **return** scanAbs($s, c_1, m, v$)

4     **if** $p$ is of type `relative difference`, $p = P_{m,v}(c_1\ X, c_2\ Y)$ **then**

5         **return** scanRel($s, c_1, c_2, m, v$)

6     **return** None     // Additional predicate classes would necessitate additional cases

7 **Func** *scanAbs(State s, Predicate p, Type c, Member m, Value v) → set[GroundPredicate]*

8     set[GroundPredicate] facts = {}     // Initialize empty set

9     **for** $i \in \{i \mid s_i[m] = v\}$ **do**

10         **if** $class(s, i) = c$ **then**

11             $g$ = GroundPredicate($p, [s_i]$)

12             facts.insert($g$)

13     **return** facts

14 **Func** *scanRel(State s, Predicate p, Type $c_1$, Type $c_2$, Member m, Value v) → set[GroundPredicate]*

15     set[GroundPredicate] facts = {}     // Initialize empty set

16     **for** $i \in \{i \mid class(s, i) = c_1\}$ **do**

17         $v_1 = s_i[m]$

18         **for** $j \in \{j \mid s_j[m] = v + v_1\}$ **do**

19             **if** $class(s, j) = c_2$ **then**

20                 $g$ = GroundPredicate($p, [s_i, s_j]$)

21                 facts.insert($g$)

22     **return** facts

---

## C  ENVIRONMENT DETAILS

This section includes more details about each environment. As mentioned in the main text, some are taken from (or based on environments from) Stella & Loguinov (2024). Example states are shown in Figs. 7, 8, 9, and 10. As an additional example of the object-oriented state representation, Fig. 6 shows the initial state from Fig. 1 in the form that an agent receives it (i.e., with no meaningful labels). Learning solely from this numerical data poses a significant challenge.

| object | type | pos | status | locked |
|--------|--------|--------|--------|--------|
| 2 | wall | (2, 0) | - | - |
| 16 | wall | (4, 2) | - | - |
| 30 | wall | (3, 5) | - | - |
| 50 | player | (5, 2) | - | - |
| 51 | door | (3, 4) | - | true |
| 52 | key | (4, 1) | free | - |
| 53 | goal | (2, 4) | - | - |

(a) human-readable object list

| object | type | attr 0 | attr 1 | attr 2 |
|--------|------|--------|--------|--------|
| 2 | 0 | (2, 0) | - | - |
| 16 | 0 | (4, 2) | - | - |
| 30 | 0 | (3, 5) | - | - |
| 50 | 1 | (5, 2) | - | - |
| 51 | 2 | (3, 4) | - | (0) |
| 52 | 3 | (4, 1) | (0) | - |
| 53 | 4 | (2, 4) | - | - |

(b) object list as observed by an agent

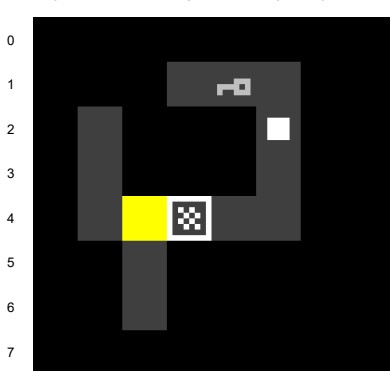

(c) a state in the `keys` domain

Figure 6: The initial state from Figure 1, now showing in (b) the state data (object list) in the form that the agent receives it (i.e., with no semantic labeling of any attribute types or values). Note that for brevity, the lists show only a subset of the objects in the state.

**Walls**   Shown in Fig. 7a, this is a maze-like domain in which the agent must learn basic relational rules. The environment's four actions allow the player to move their character by one unit in each of the four grid directions. If the character would move into a wall, the action does nothing. Although this seems simple to humans, learning the rules of this environment directly from object-oriented transitions is difficult for existing methods. Notably, the importance of local rules (i.e., checking for walls near the player) is not included in the information that the agent receives.

**Doors**   Shown in Fig. 7b, this domain extends `walls` with the addition of door objects and a color attribute. Both the player character and the doors possess the color attribute, which takes values in $\{0, 1\}$ in this domain. The agent can use the new `change-color` action toggle its color. Doors that are a different color from the player block its movement. Thus, the rules for this domain are more complex than those of the `walls` domain.

**Fish**   Shown in Fig. 7c, this domain replaces the player character with one or more fish objects that move in a random direction – conditional on the surrounding walls – at each step. Thus, this environment tests an agent's ability to robustly model stochastic transition functions.

**Gates**   Shown in Fig. 7d, this is a highly-complex grid-world environment that features a large number of classes and actions. The new gate objects block normal player movement, except that the

player can jump over gates (as long as the other side is not blocked) using the new `jump` actions (one for each direction). The guard object, which is controlled independently of the player, is not blocked by gates. Switches are also spread randomly throughout the walkable parts of the level; whenever the player moves over a switch, its state is toggled.

**Maze**    Shown in Figs. 8a and 8b, this environment augments the `walls` domain with a reward signal and goal objects. By default, all actions receive a penalty of $-1$ to incentivize the agent to take the shortest path to a goal. Actions that attempt to move the player into a wall instead receive a penalty of $-2$, since the agent should never take such an action. If the agent does not attempt to move into a wall, and its action results in it standing on a goal (i.e., by moving onto a goal or by choosing to stay still when already on a goal), it receives a reward of $+1$. Although these dynamics seem simple to most humans, we find that existing algorithms cannot learn this domain.

**Coins**    Shown in Figs. 8c and 8d, this environment is similar to the `maze` domain, but it replaces the goals with coins. Unlike goals, coins disappear (i.e., are "picked up") when the agent moves over them, so each only gives a single reward. Thus, this domain encodes a routing problem, similar to the Traveling Salesman Problem.

**Keys**    Shown in Figs. 9a and 9b, this environment adds keys and doors to the `maze` domain. Doors are initially locked, preventing the player from passing through them. To move through a door, the player must unlock it by bringing an unused key to it. Although any key can be used to open any door, each key can only be used once. Out of all of the tested domains, this is the most challenging to learn, likely due to the presence of highly complex and rare interactions (e.g., the player cannot step onto an unused key if it is currently holding another key).

**Lights**    Shown in Fig. 9c, this is a simple non-grid-world domain in which the agent controls a tunable remote that can be used to toggle lights. We augment the version from Stella & Loguinov (2024) with a reward signal such that the agent receives a small penalty for tuning the remote (decrementing or incrementing the channel), a large penalty for turning a light on, and a large reward for turning a light off.

**Switches**    Shown in 9d, this domain combines `walls` and `lights` into a more complex and interesting scenario. Here, the player moves around a maze filled with switches and lights (formed in pairs, indicated in our example images by their hue). When on top of a switch, the agent can take the `toggle` action to mutate the state of the corresponding light, regardless of the position of the light in the level. Thus, unlike the other grid-world domains, the `switches` environment contains a kind of non-local behavior.

**Scale($n_p, n_c$)**    Shown in Fig. 10, this set of environments augments the `walls` domain with $n_p$ distinct player classes, each of which has $n_c$ copies of each movement action. This allows us to evaluate the way an algorithm's learning speed scales with the number of classes and actions in a manner that keeps all else (i.e., the complexity of the environment's rules) equal. In the example images, since players may overlap, each player is indicated by a dot in a different position within the white squares.

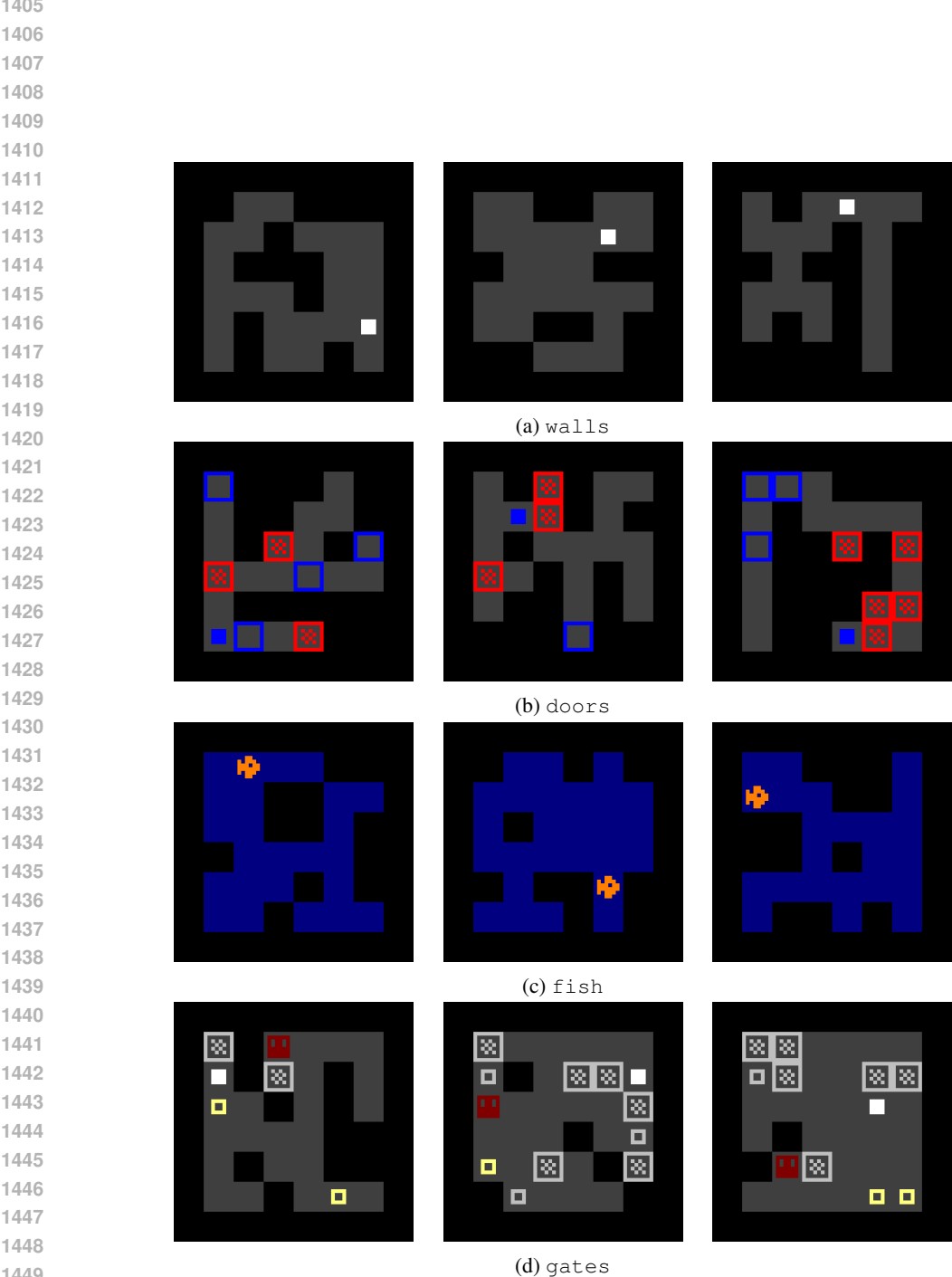

(a) walls

(b) doors

(c) fish

(d) gates

Figure 7: Example states, pt. 1 (walls, doors, fish, gates)

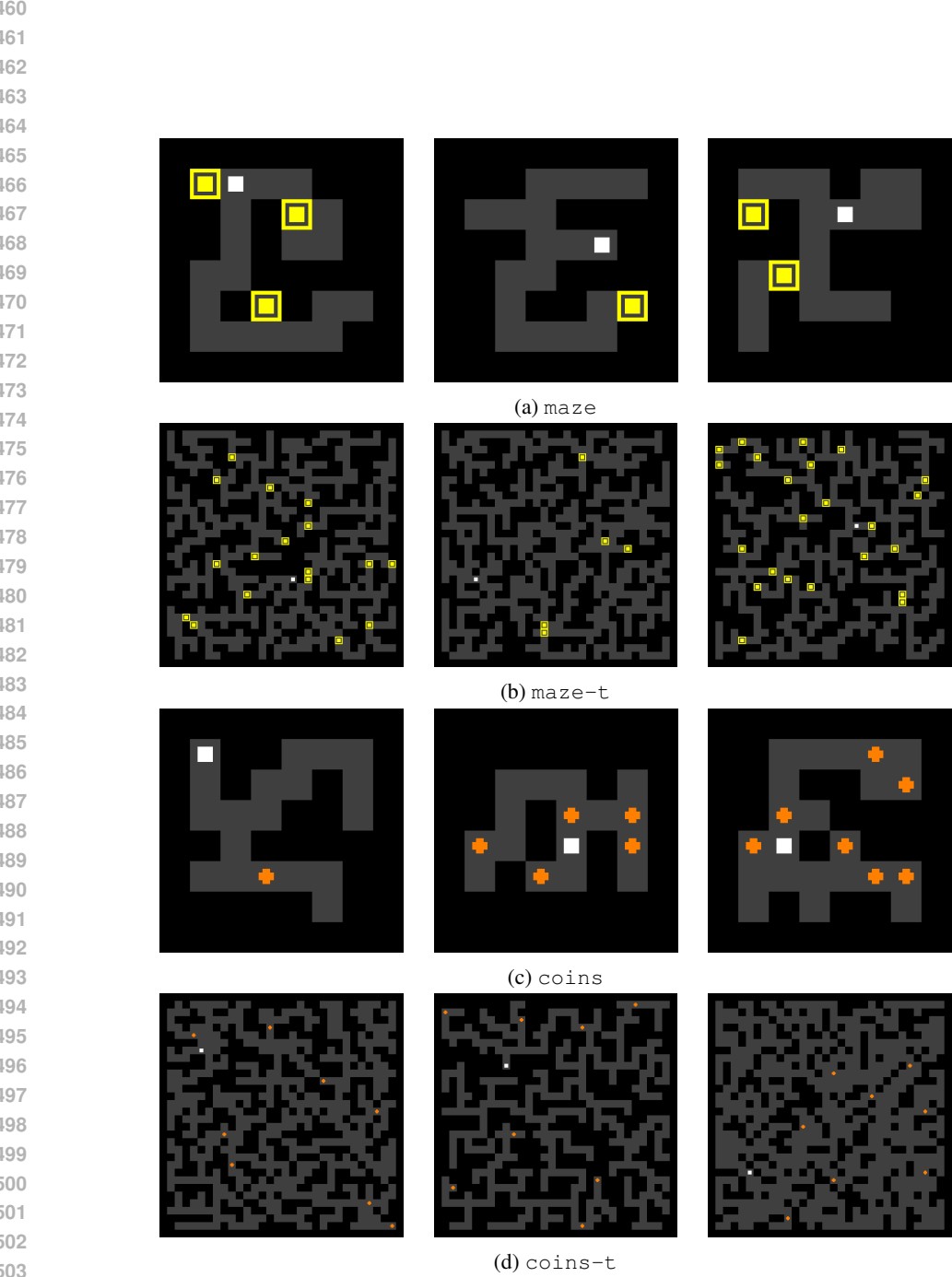

(a) `maze`

(b) `maze-t`

(c) `coins`

(d) `coins-t`

Figure 8: Example states, pt. 2 (`maze`, `maze-t`, `coins`, `coins-t`)

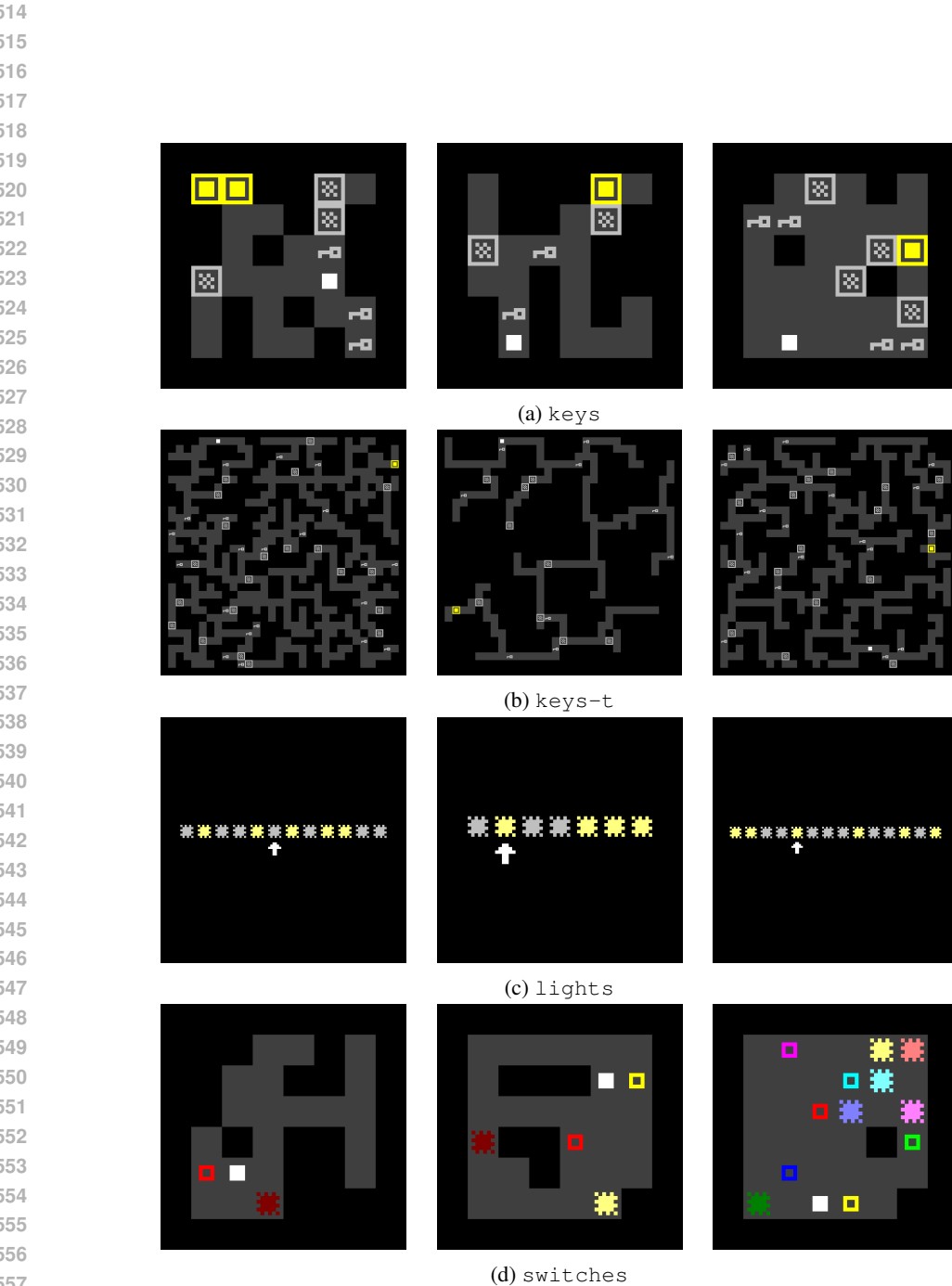

(a) `keys`

(b) `keys-t`

(c) `lights`

(d) `switches`

Figure 9: Example states, pt. 3 (`keys`, `keys-t`, `lights`, `switches`)

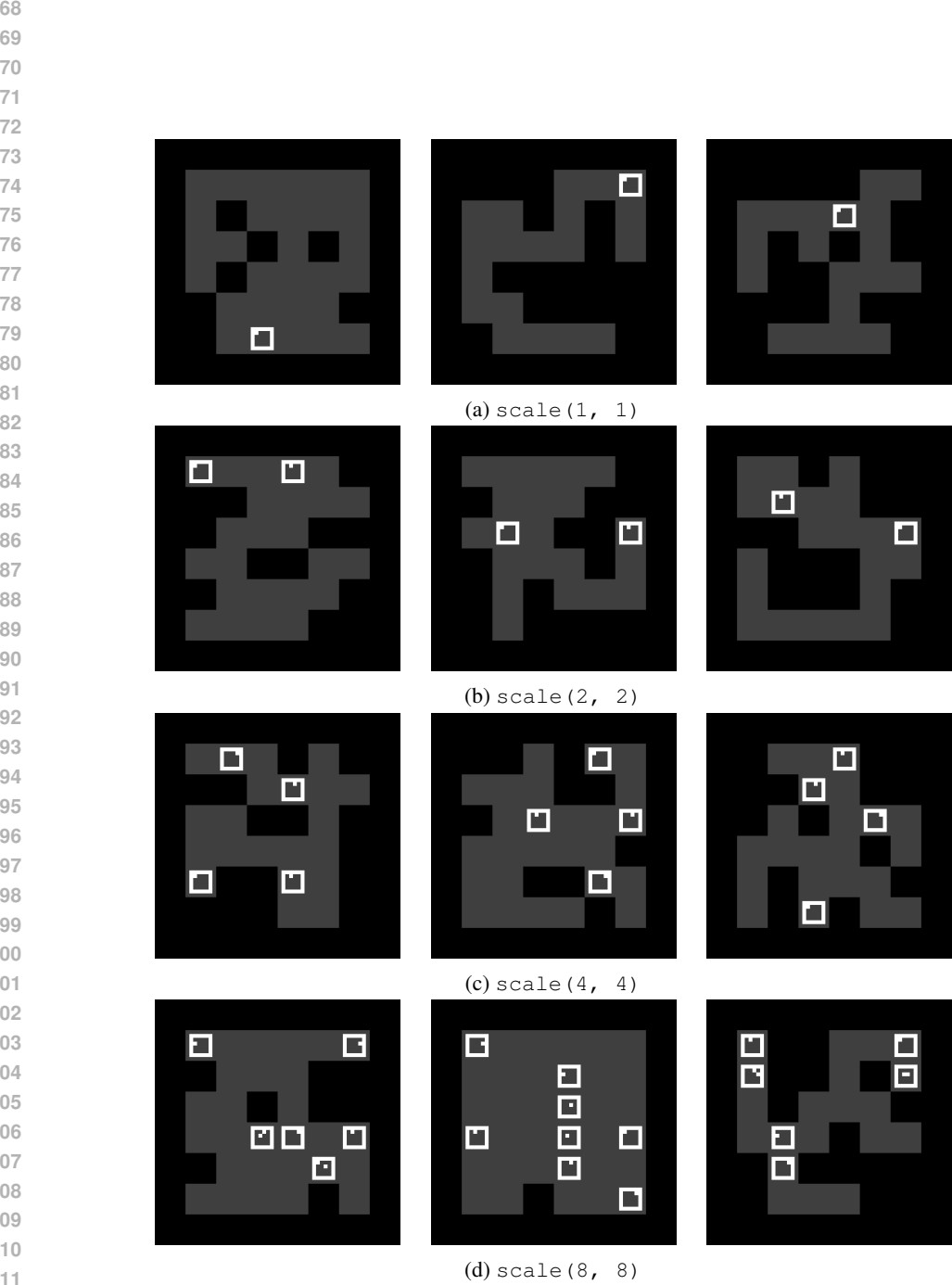

(a) `scale(1, 1)`

(b) `scale(2, 2)`

(c) `scale(4, 4)`

(d) `scale(8, 8)`

Figure 10: Example states, $\texttt{scale}(n_p, n_c)$

## D  EXPERIMENT DETAILS: TREETHINK VS. QORA

In each experiment (except those where QORA failed to complete training due to, e.g., causing the test machine to crash), we run both TreeThink and QORA ten times, averaging the runs' results together. We used the same $\alpha$ setting for both algorithms within each experiment, $\alpha = 0.01$ unless otherwise stated. Results are shown in Figs. 11, 12, 13, 14, 15, 16, 17, and 18. Note that because several machines were used for testing, timing results are not always comparable across different domains or configurations. However, each row of plots corresponds to a single test, so timings can be compared within the row.

Figs. 11, 12, 13, and 14 show the full data collected from tests in each domain (other than the scaling tests). Fig. 15 shows transfer tests (maze-t, coins-t, and keys-t). Figs. 16, 17, and 18 show results from the $\texttt{scale}(n_p, n_c)$ domains.

Several examples of trees constructed by TreeThink are shown in Fig. 19.

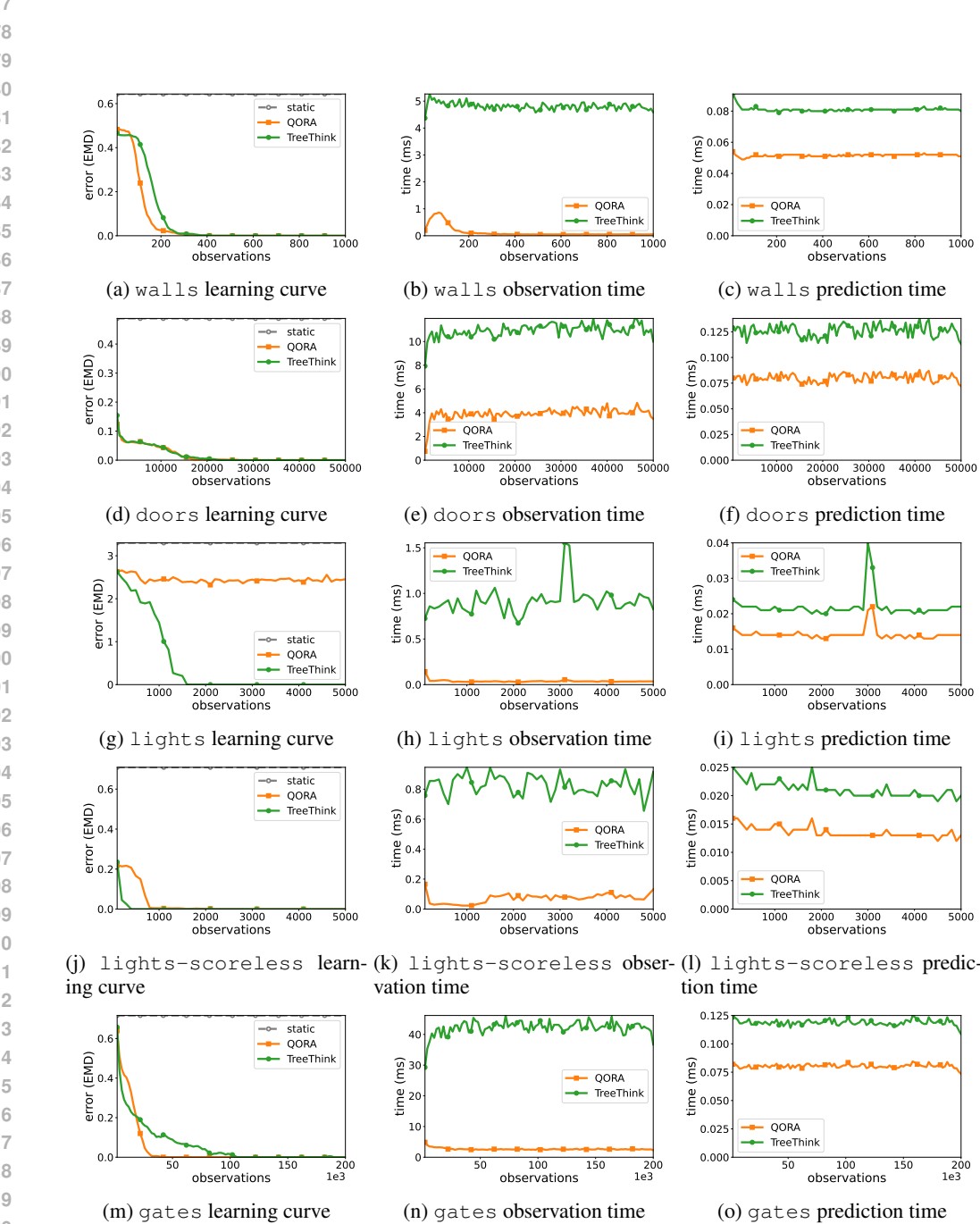

(a) walls learning curve

(b) walls observation time

(c) walls prediction time

(d) doors learning curve

(e) doors observation time

(f) doors prediction time

(g) lights learning curve

(h) lights observation time

(i) lights prediction time

(j) lights-scoreless learning curve

(k) lights-scoreless observation time

(l) lights-scoreless prediction time

(m) gates learning curve

(n) gates observation time

(o) gates prediction time

Figure 11: TreeThink vs. QORA (walls, doors, lights, lights-scoreless, gates)

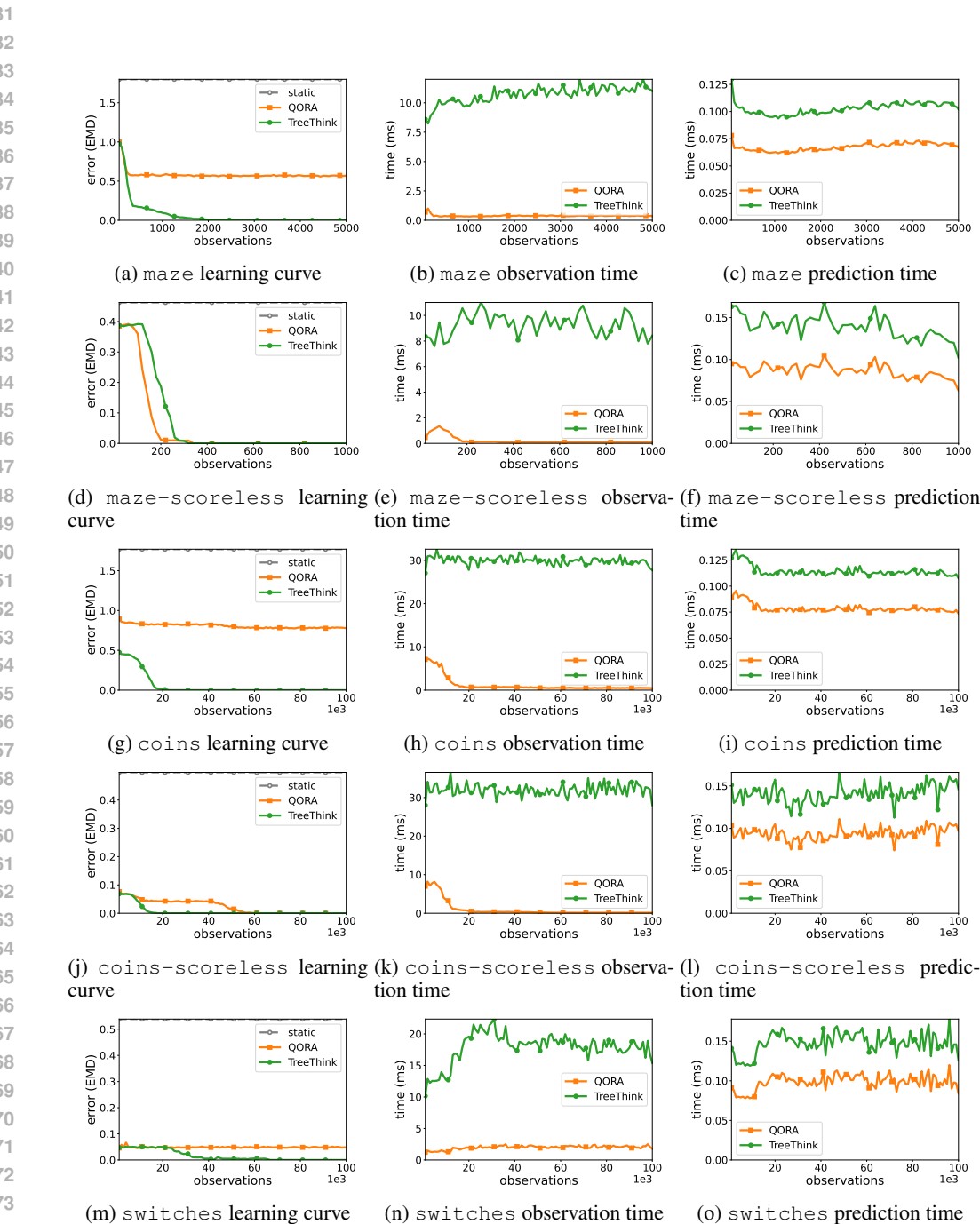

Figure 12: TreeThink vs. QORA (`maze`, `maze-scoreless`, `coins`, `coins-scoreless`, `switches`)

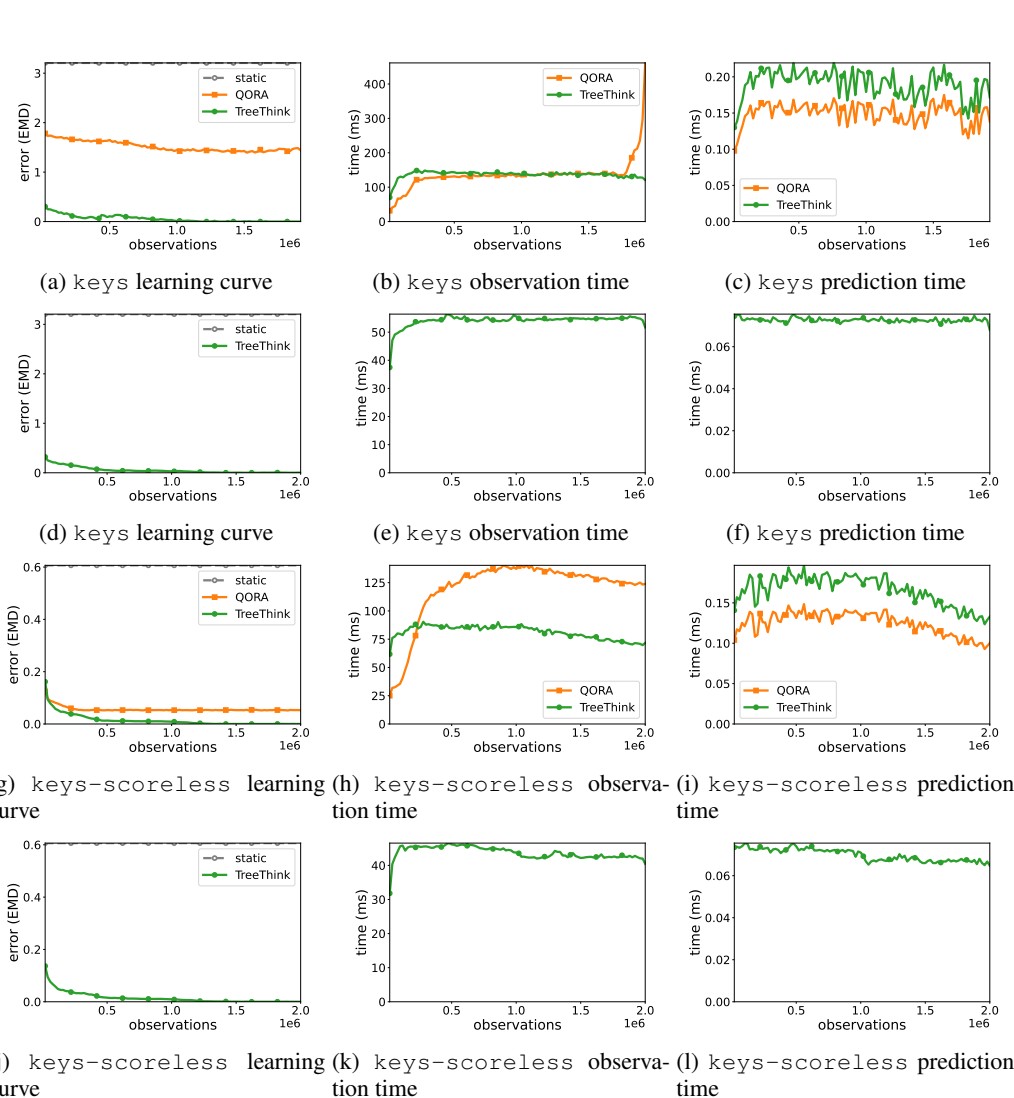

Figure 13: TreeThink vs. QORA (`keys` and `keys-scoreless`); since QORA could not finish in some cases, we also ran experiments with just TreeThink.

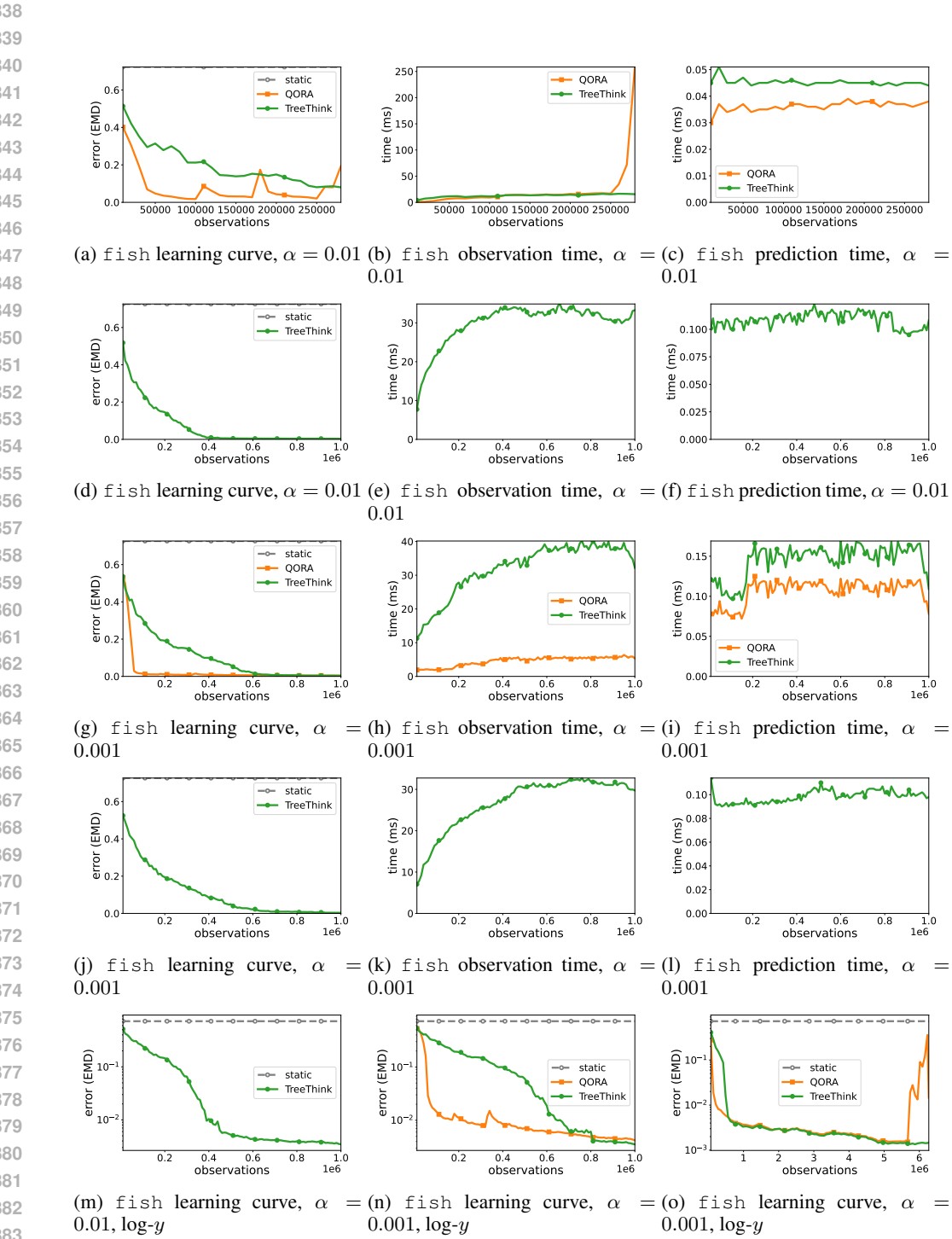

(a) fish learning curve, $\alpha = 0.01$

(b) fish observation time, $\alpha = 0.01$

(c) fish prediction time, $\alpha = 0.01$

(d) fish learning curve, $\alpha = 0.01$

(e) fish observation time, $\alpha = 0.01$

(f) fish prediction time, $\alpha = 0.01$

(g) fish learning curve, $\alpha = 0.001$

(h) fish observation time, $\alpha = 0.001$

(i) fish prediction time, $\alpha = 0.001$

(j) fish learning curve, $\alpha = 0.001$

(k) fish observation time, $\alpha = 0.001$

(l) fish prediction time, $\alpha = 0.001$

(m) fish learning curve, $\alpha = 0.01$, log-$y$

(n) fish learning curve, $\alpha = 0.001$, log-$y$

(o) fish learning curve, $\alpha = 0.001$, log-$y$

Figure 14: TreeThink vs. QORA (fish); since QORA could not finish in some cases, we also ran experiments with just TreeThink. Semilog-$y$ plots are included to better visualize small values.

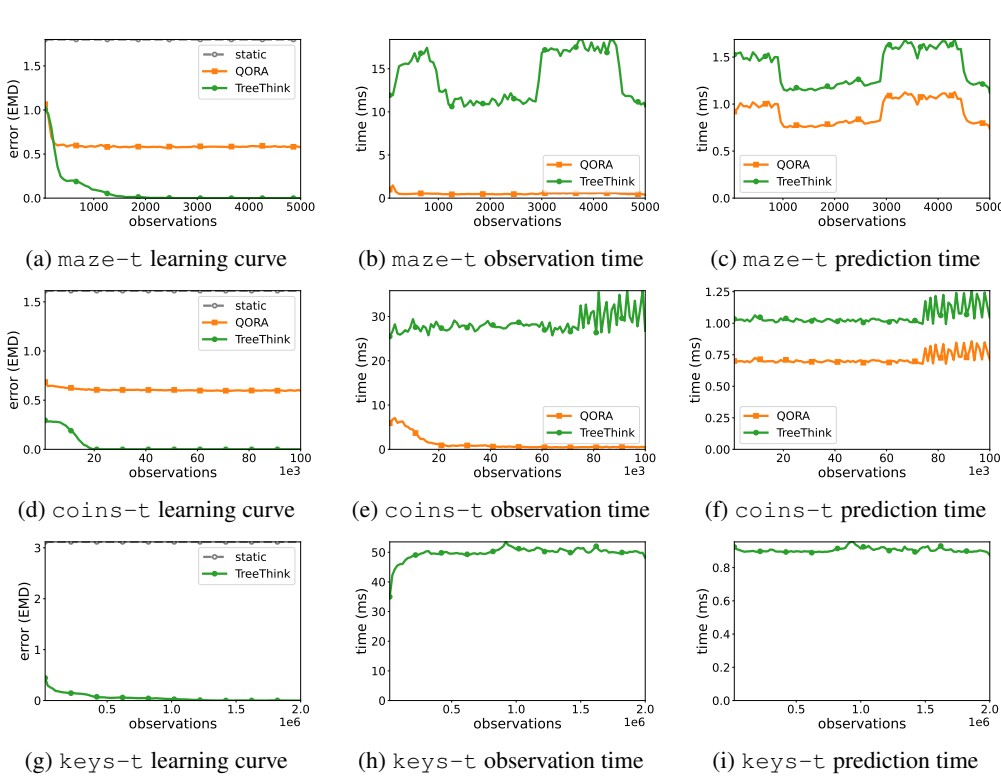

(a) maze-t learning curve

(b) maze-t observation time

(c) maze-t prediction time

(d) coins-t learning curve

(e) coins-t observation time

(f) coins-t prediction time

(g) keys-t learning curve

(h) keys-t observation time

(i) keys-t prediction time

Figure 15: TreeThink vs. QORA, transfer to larger instances ($32 \times 32$) while training in smaller worlds ($8 \times 8$). We ran only TreeThink in the keys domain because QORA often crashes. TreeThink displays perfect generalization in each environment.

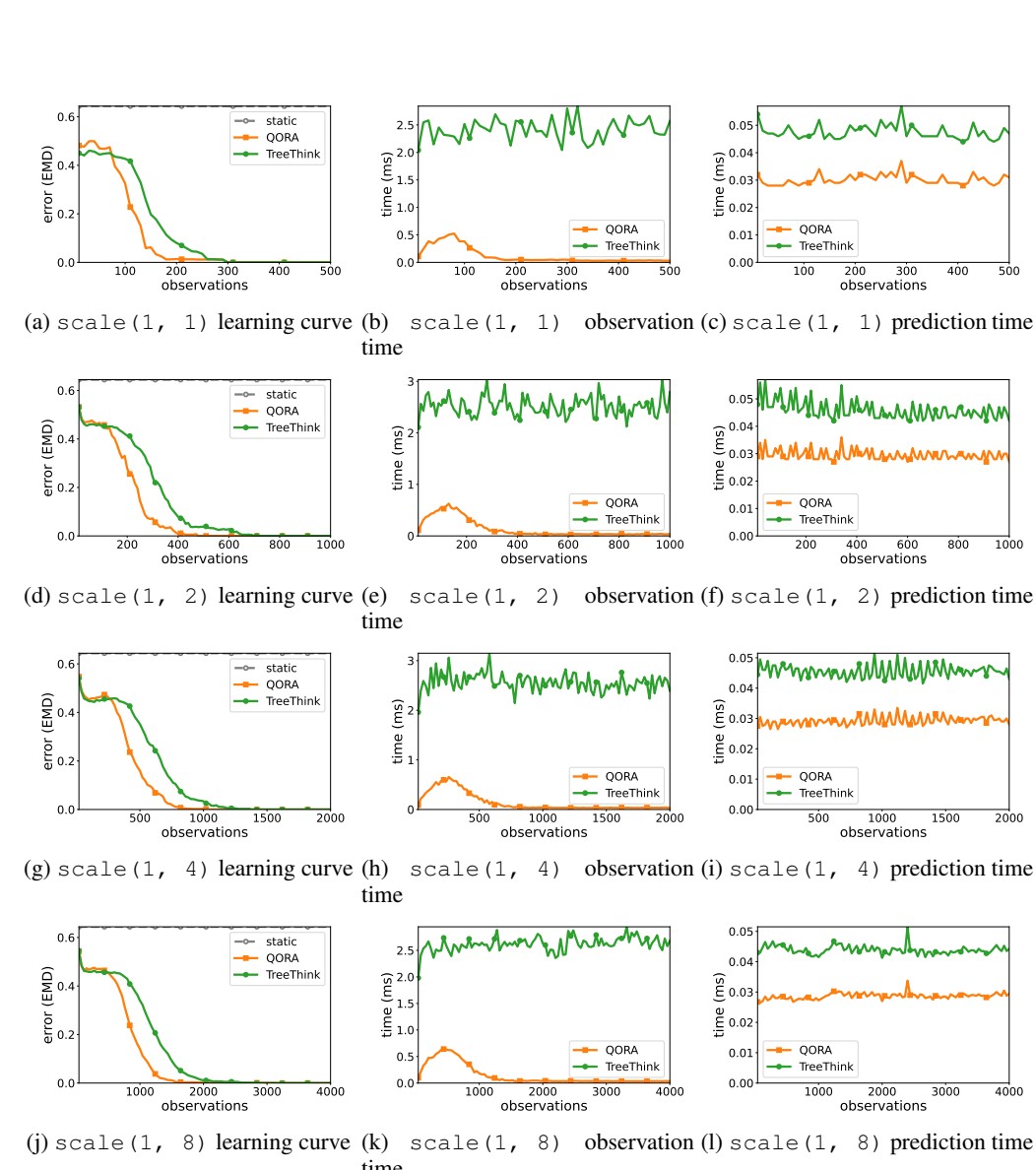

(a) scale(1, 1) learning curve (b) scale(1, 1) observation time (c) scale(1, 1) prediction time

(d) scale(1, 2) learning curve (e) scale(1, 2) observation time (f) scale(1, 2) prediction time

(g) scale(1, 4) learning curve (h) scale(1, 4) observation time (i) scale(1, 4) prediction time

(j) scale(1, 8) learning curve (k) scale(1, 8) observation time (l) scale(1, 8) prediction time

Figure 16: TreeThink vs. QORA, scaling tests: $n_c = 1$, varying $n_p$. Note that as the $x$-axis scales up proportionally to $n_p$, the plots maintain the same proportions, meaning that learning time is scaling up linearly with the number of classes (and corresponding actions).

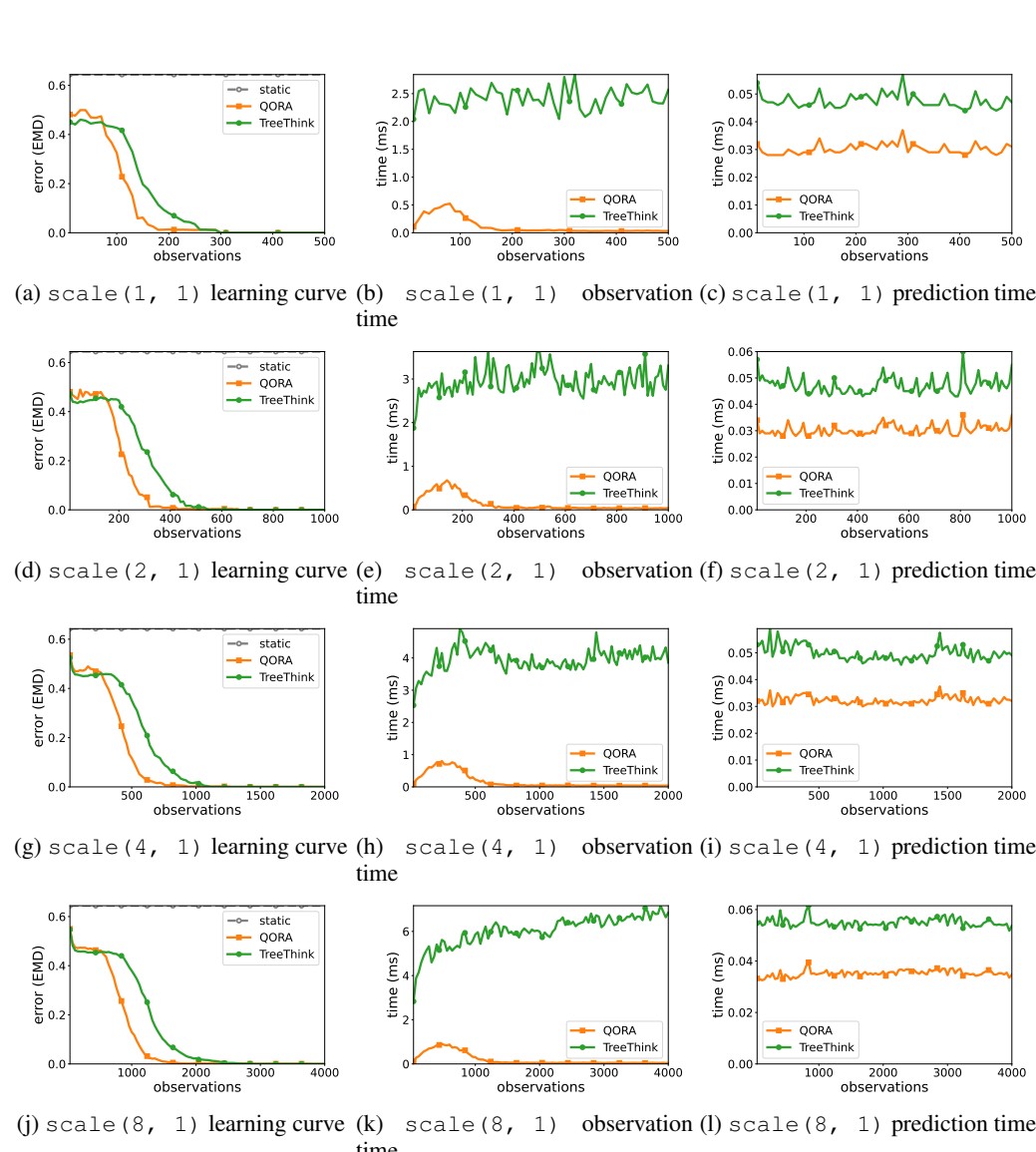

Figure 17: TreeThink vs. QORA, scaling tests: varying $n_c$, $n_p = 1$. Note that as the $x$-axis scales up proportionally to $n_c$, the plots maintain the same proportions, meaning that learning time is scaling up linearly with the number of actions.

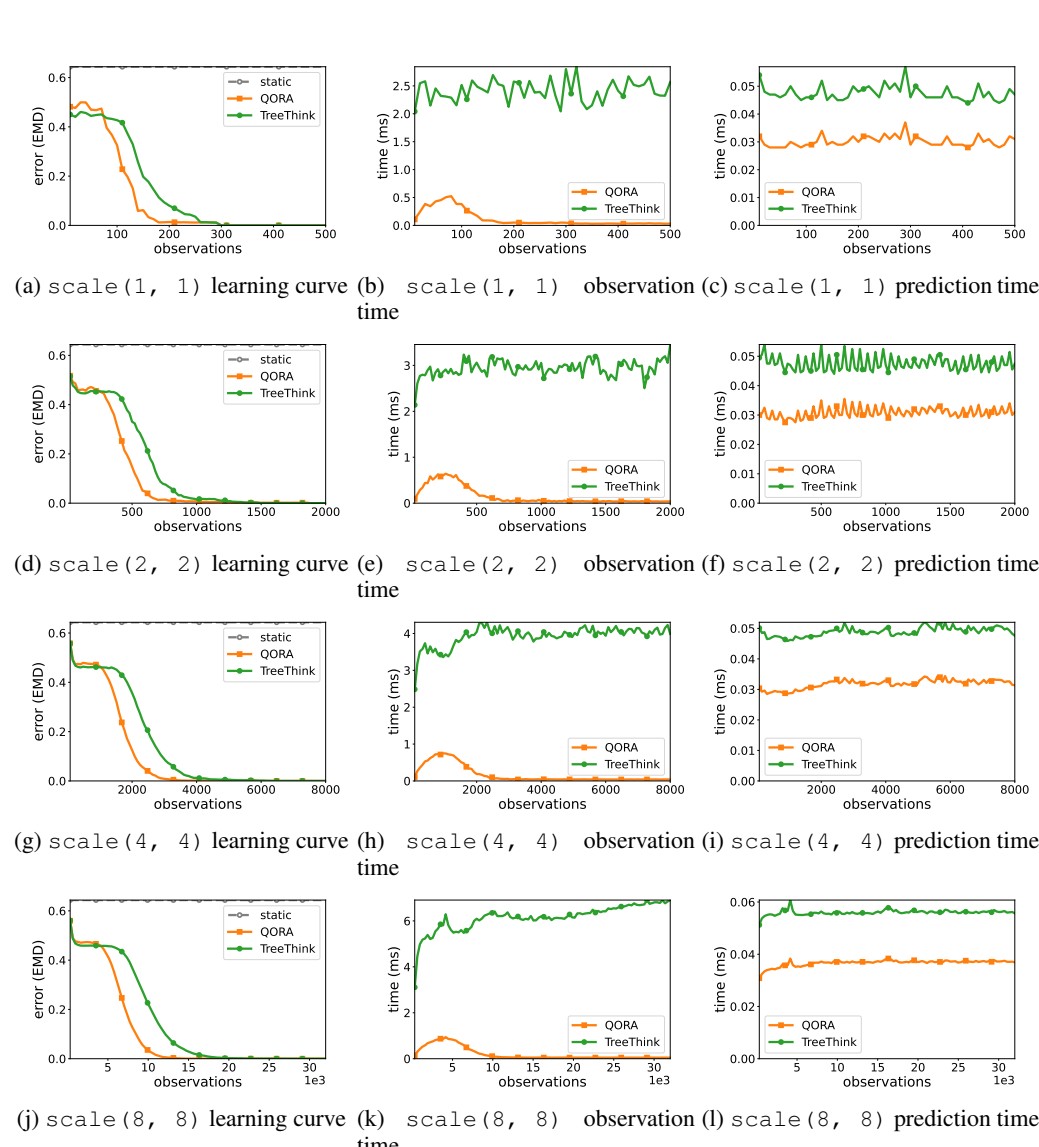

(a) scale(1, 1) learning curve (b) scale(1, 1) observation time

(c) scale(1, 1) prediction time

(d) scale(2, 2) learning curve (e) scale(2, 2) observation time

(f) scale(2, 2) prediction time

(g) scale(4, 4) learning curve (h) scale(4, 4) observation time

(i) scale(4, 4) prediction time

(j) scale(8, 8) learning curve (k) scale(8, 8) observation time

(l) scale(8, 8) prediction time

Figure 18: TreeThink vs. QORA, scaling tests: varying $n_c$ and $n_p$. Note that as the $x$-axis scales up proportionally to $n_c \times n_p$, the plots maintain the same proportions, meaning that learning time is scaling up linearly with the number of actions.

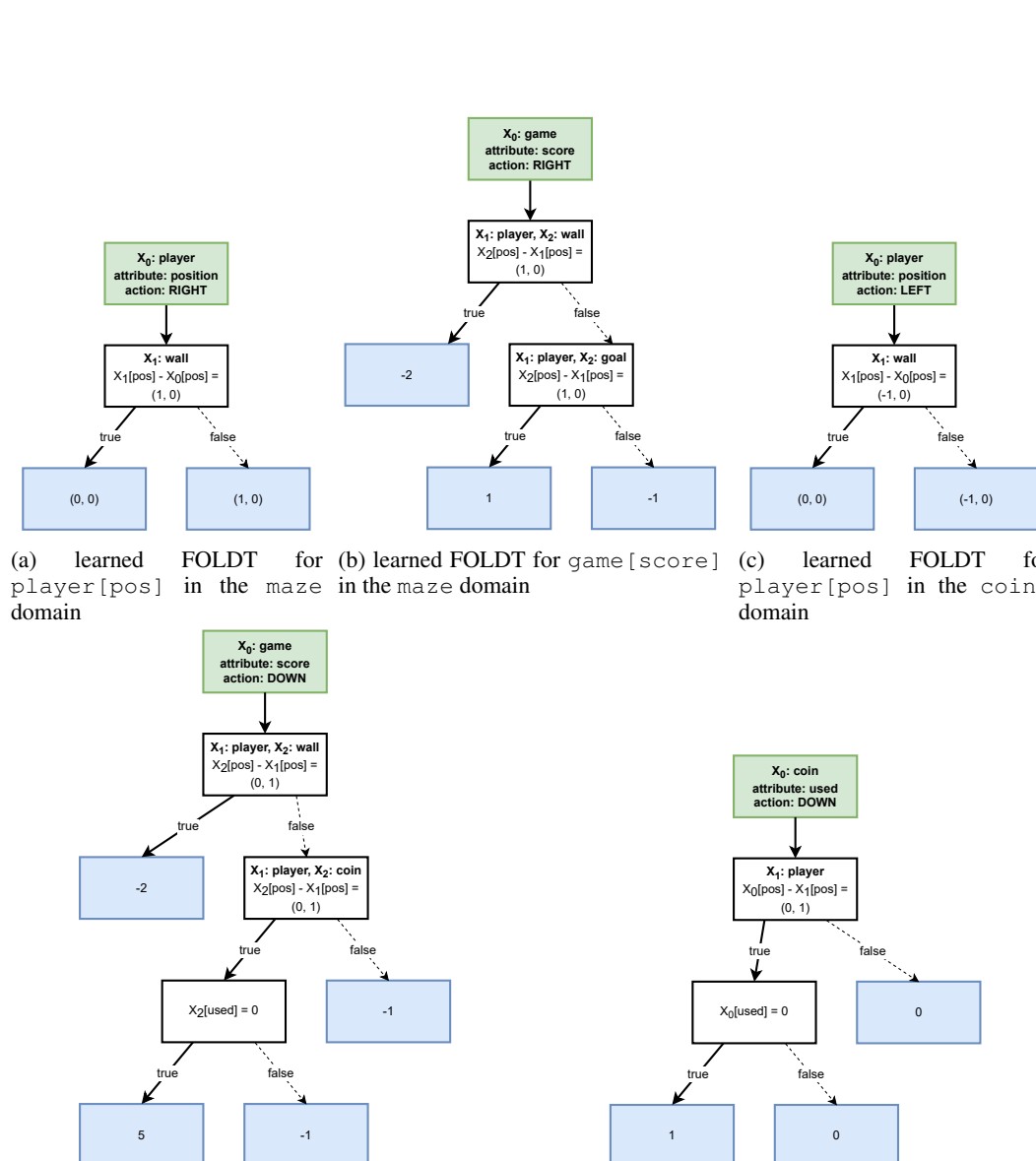

(a) learned FOLDT for `player[pos]` in the `maze` domain

(b) learned FOLDT for `game[score]` in the `maze` domain

(c) learned FOLDT for `player[pos]` in the `coins` domain

(d) learned FOLDT for `game[score]` in the `coins` domain

(e) learned FOLDT for `coin[used]` in the `coins` domain

Figure 19: Example FOLDTs learned by TreeThink in the `maze` and `coins` domains. The top (green) box of each tree notes the tree's $(c, m, a)$ triplet and its argument variable $X_0$. Variables bound by subsequent branch tests are labeled at the top of the corresponding box (e.g., in (c), $X_1$: wall). Note that if a test fails (i.e., the right branch is taken), its variables are not bound; hence, in the second test in tree (b), $X_1$ and $X_2$ are not related to the $X_1$ and $X_2$ from the prior test.

# E EXPERIMENT DETAILS: ABLATION TESTS

Since the inference time stays consistent over time, we report the average time per `predict` call from each set of runs. Figure 20 shows both a chart, to visualize the massive performance improvement, as well as a table, for more detailed comparison.

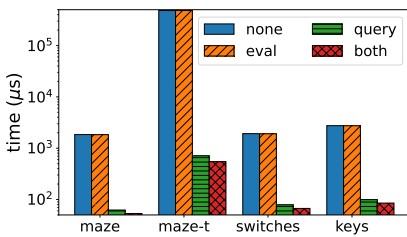

| domain | none | eval | query | both |
|---|---|---|---|---|
| maze | 1842 | 1835 | 62.3 | 53.2 |
| maze-t | 481822 | 481597 | 710 | 546 |
| switches | 1920 | 1909 | 78.6 | 67.0 |
| keys | 2748 | 2736 | 99.4 | 85.2 |

(a) average inference time ($\mu$s) chart          (b) average inference time ($\mu$s) table

Figure 20: Average inference time in four domains, varying optimizations. Settings are: none (no optimizations to inference), eval (optimizing tree evaluation), query (optimizing state queries), and both (optimizing state queries and tree evaluation).

## F   EXPERIMENT DETAILS: NEURAL BASELINES

We apply a custom NPE architecture, shown in Figure 21, based on the design that Stella & Loguinov (2024) used for object-oriented transition learning. The objects in a state are input as vectors $X_i$, formed by concatenating the object's one-hot-encoded class and all of its attribute value vectors (if an object is lacking some attribute, a zero vector of the appropriate size is substituted). The action $a \in A$ is also input to the network, one-hot encoded.

The blocks $F_1$ and $F_2$ are feed-forward networks comprising alternating linear and ReLU layers ( $F_1$ ends with an ReLU layer, $F_2$ ends with a linear layer). Let $d$ be the total length of an object vector (since they are all the same length) and $e$ be the length of the output vector of $F_1$. Then, the first layer of $F_1$ has width $2d$ and the first layer of $F_2$ has width $d + |A| + e$. To predict object attributes, the output dimension of $F_2$ is $d$.

To make the network structure simpler and more efficient, we keep the reward signal separate. A second network, which outputs a scalar value, is used to model the reward. This network uses an NPE internally, but the final sum $X_i^{(t)} + \Delta_i^{(t+1)}$ is skipped (the output of the network is just $\Delta_i^{(t+1)}$). This allows us to set $F_2$ to output a vector that is not of size $d$. The reward network sums all of the outputs of $F_2$, $\sum_{i=1}^{n_s} \Delta_i^{(t+1)}$, then passes the result through a linear layer to produce a scalar output.

Our hand-tuned networks for the `maze` environment, where $d = 5$ and $|A| = 5$, use the following network dimensions:

- $T$, $F_1$: $10 \rightarrow 16 \rightarrow 8 \rightarrow 8 \rightarrow 4$
- $T$, $F_2$: $14 \rightarrow 4 \rightarrow 5$
- $R$, $F_1$: $10 \rightarrow 16 \rightarrow 8 \rightarrow 8 \rightarrow 16$
- $R$, $F_2$: $26 \rightarrow 16$

The weights (approximately four hundred hand-tuned values) are included in the codebase that will be released upon publication. Although not shown in the plots (because it is not very interesting to look at), we included a network with the hand-tuned weights in all of our experiments. It got perfect accuracy (zero error) on all tested transitions.

For training, we run several variations of the hand-tuned architecture. We use NPE_X_Y to denote a network $X$ times wider than our hand-crafted design with $Y - 1$ extra layers in each of $F_1$ and $F_2$ (all the same width as the layer before them, i.e., the hand-crafted last layer width times $X$). The networks collect observations in episodes; one epoch of training occurs at the end of each episode. We utilize a replay buffer with one thousand slots, which is split into ten random batches for each epoch. While our initial experiments used a typical deque replay buffer, this approach was unsuccessful; to increase the variety in the data, we moved to a replay buffer that, when full, randomly selects an existing item to evict. This led to the results we show in the paper. After trying several optimizers and hyperparameter values, we settled on AdamW (supplied by PyTorch) with a learning rate of $0.001$.

Additional results from our NPE experiments are shown in Figure 22.

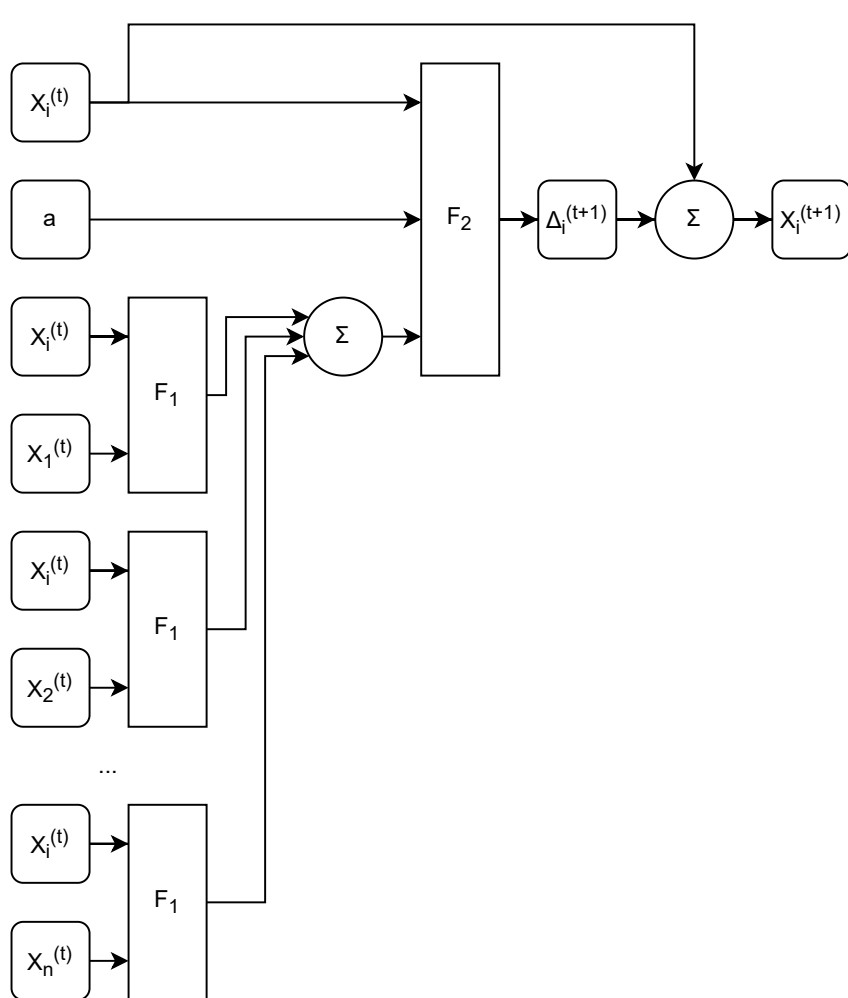

Figure 21: The architecture used for our NPE implementation, based on the structure described by Stella & Loguinov (2024) for object-oriented transition learning. The $F_1$ and $F_2$ modules are feed-forward networks comprising alternating linear and ReLU layers.

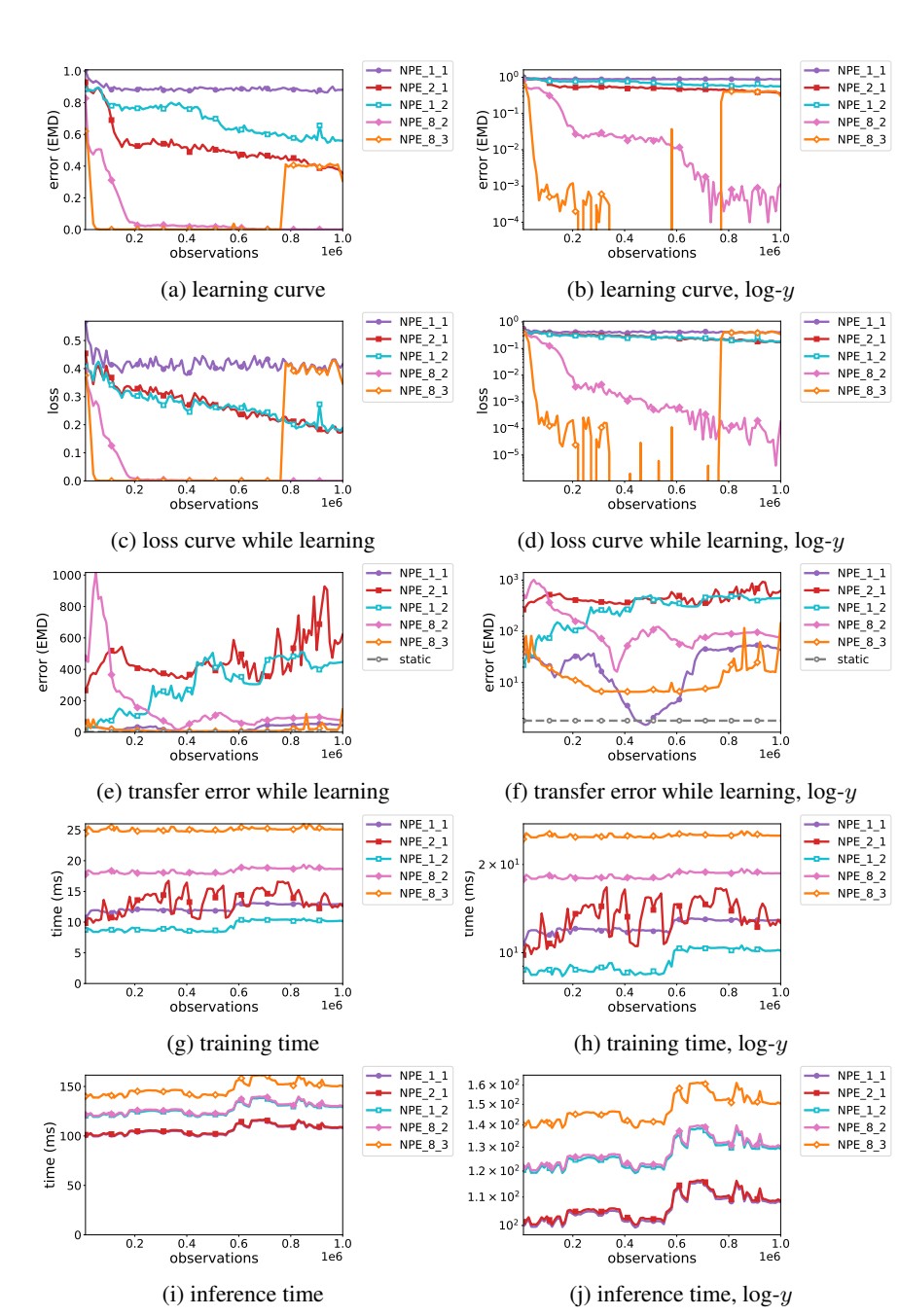

Figure 22: Using various NPE architectures to model the `maze` domain. Semilog-$y$ plots are included to better visualize small values. Breaks in a line (e.g., the error of NPE_8_3) in semilog-$y$ plots indicate zeroes.

## G  PLANNING EXPERIMENTS

We now show results of planning experiments using Monte-Carlo Tree Search (MCTS) with the pUCT rule (Coulom, 2007; Rosin, 2010; Schrittwieser et al., 2020) and a computation budget of 100 simulations (i.e., model evaluations) per action. Scores are normalized for each episode into the range $[-1, 1]$, where $-1$ indicates *pessimal* performance (the lowest score obtainable for that episode), $0$ indicates *trivial* performance (that of an agent that does nothing, neither productive or harmful), and $1$ indicates *optimal* performance. The various settings shown in our experiments, which vary the level size, number of walls, number of goals, and episode length, are detailed in Table 1.

Table 1: Planning experiment settings

| Width | Height | Interior walls | Goals | Episode length |
|---|---|---|---|---|
| 8 | 8 | 10 | 2 | 10 |
| 10 | 10 | 20 | 5 | 20 |
| 12 | 12 | 50 | 10 | 30 |
| 14 | 14 | 80 | 15 | 40 |
| 16 | 16 | 160 | 20 | 50 |

Our planner uses solely an environment model (supplying $\hat{T}$ and $\hat{R}$); the prior policy is uniform (i.e., no action is given any special weight *a priori* during search) and the value estimator outputs zero for all states. Nonetheless, as shown in Figure 23, the planner is able to get near-optimal scores using both TreeThink (trained in the same way as in our prior experiments, i.e., in $8 \times 8$ levels) and NPE* (our hand-tuned perfectly-accurate neural network).

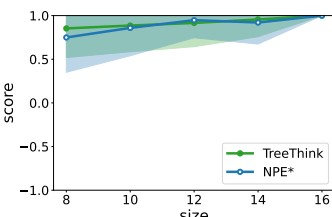

Figure 23: TreeThink (fully trained) and NPE* (hand-tuned to be perfectly accurate) planning in various world sizes; highlights show one standard deviation

We next take NPE_1_1 and NPE_8_3 and train them using the same setup as in Section 3.4 and Appendix F. During training, we periodically (every 100k observations) run the MCTS planner using the partially-trained models. The results are shown in Figure 24. The different behavior between NPE_1_1 and NPE_8_3, especially as the levels become more complex, is immediately apparent. While NPE_1_1 never achieves perfect accuracy (see Figure 22 in Appendix F), it also seems to overfit less; thus, although it never performs well, MCTS with NPE_1_1 is much more stable as level size increases. On the other hand, NPE_8_3 is able to (for some amount of time) accurately model the $8 \times 8$ levels, allowing MCTS to achieve relatively high scores in this setting. However, the model apparently *drastically* overfits, which leads MCTS to find low-quality plans in the other settings. In fact, even in $12 \times 12$ levels, the planner using NPE_8_3 begins returning plans that are *almost as bad as possible*. In real-world deployment, this kind of outcome – where the agent suddenly takes harmful actions upon transfer to new conditions – could be extremely dangerous.

For further analysis, Figure 25 shows the performance of each model checkpoint across each level size. Again, NPE_1_1 remains stable, even as training progresses. In contrast, while NPE_8_3 initially (@0) gets similar performance across all level sizes, but after even just a small amount of training, it overfits to the $8 \times 8$ worlds. Further training improves performance in this setting in exchange for poor returns in all other scenarios.

Finally, Figure 26 compares the planning time of MCTS using TreeThink, NPE*, NPE_1_1, and NPE_8_3. As expected, the runtime using NPE* and NPE_1_1 are essentially identical, since the

architectures are the same (only the parameters differ). Notably, the planner using TreeThink is by far the fastest due to the lower time required to evaluate the TreeThink models.

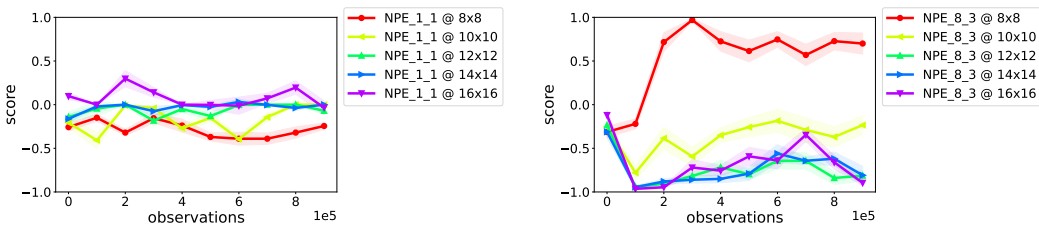

Figure 24: Planning in the maze domain with NPE_1_1 and NPE_8_3 during training, varying world sizes. Highlights show $1/4$ stdev, to preserve clarity.

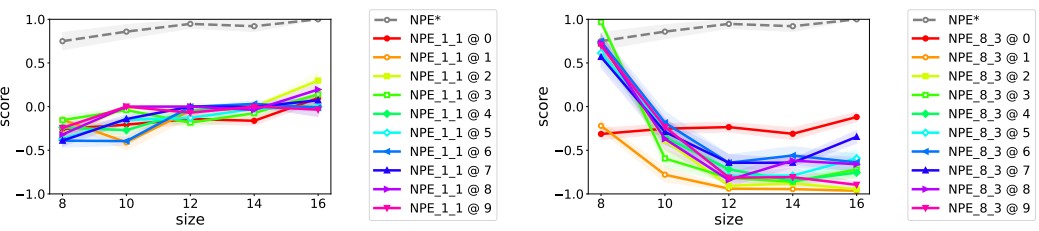

Figure 25: Planning in the maze domain with NPE_1_1 and NPE_8_3 across several world sizes, varying number of observations. Highlights show $1/4$ stdev, to preserve clarity.

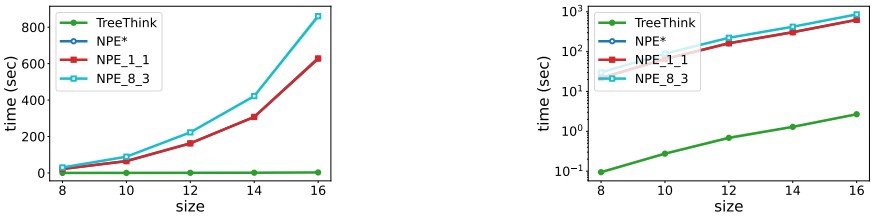

(a) planning time per episode as world size increases

(b) planning time per episode as world size increases, log-$y$

Figure 26: Planning time in the maze domain.

