# OpenReview forum: "Object-Oriented Transition Modeling with Inductive Logic Programming"
_ICLR.cc/2026/Conference — Submitted to ICLR 2026_

### Official Review · Reviewer_vpUZ · 2025-10-20

**Soundness:** 2
**Presentation:** 2
**Contribution:** 2
**Rating:** 2
**Confidence:** 4

**Summary:**

The paper essentially trains a "reinforcement learning" agent (although the authors never use this word). Unlike other approaches (e.g., deep reinforcement learning), the agent employs first-order logical decision trees. A particular contribuiton is the facilitation of online learning/updating of the tree.

**Strengths:**

- Very interesting and original idea to use first-order logical decision trees as a reinforcement learning agent
- The learned trees are interpretable
- The approach outperforms the previous approach (Stella & Loguinov 2024)

**Weaknesses:**

- The learning framework / task is not formally defined. Section 2 directly starts with an algorithm before clearly defining the problem setting.
- The paper seems closely related to reinforcement learning. However, the authors do not use the term reinforcement learning and do not discuss how the paper is similar/dissimilar. This makes the paper rather confusing.
- The approach is applied to toy problems like a robot traversing a maze. The practical/scientific relevance does not become fully clear.
- The TreeLearning algorithm makes sense. However, it does not seem to be particularly challenging. Overall, without a clear problem definition and without a clear algorithmic contribution, I cannot see a significant contribution of this paper.

**Questions:**

- How would you formally define the learning task?
- How is your approach related to reinforcement learning?
- How big are the generated trees (in number of nodes)? Are they easily interpretable by a human?
- Why do you not use a classical reinforcement learning approach as a baseline? (e.g., in Figure 3)

---

### Official Review · Reviewer_YQ9K · 2025-10-28

**Soundness:** 2
**Presentation:** 2
**Contribution:** 2
**Rating:** 2
**Confidence:** 3

**Summary:**

The paper proposes TreeLearn/TreeThink, an inductive-logic-programming (ILP) approach to object-oriented transition learning. States are sets of typed objects with attributes; for each action, the method learns a collection of independent per-attribute rules represented as first-order logical decision trees. Prediction iterates over all objects/attributes and applies the corresponding rule; leaves return deltas that are added to current values. The reward is folded into the state via a special game[score] object that tracks the cumulative sum of rewards, so no separate reward model is learned. Experiments across grid-world-style domains compare TreeThink to QORA and neural baselines, claiming faster convergence, stability, and transfer to larger layouts.

**Strengths:**

- Interpretable model class learning (FOLDT)
- Object centric nature

**Weaknesses:**

- Method is described too abstractly in the main text; intuition and operational details are hard to extract
- Strong modeling assumption: attributes are predicted independently; the paper does not justify, analyze, or ablate this restriction.
- Reward modeling: including cumulative reward as part of state breaks the standard MDP Markov assumption unless extra conditions hold; this is neither acknowledged nor analyzed.
- related work is thin given a long history in action-model learning, model-based RL, and relational/OO RL; the “object-oriented vs. relational” distinction is asserted but not clarified, and in practice they overlap.

**Questions:**

1. Lack of intuition and clarity
The paper’s exposition focuses on what TreeThink does rather than why or how it works.
Sections describing the algorithm list steps such as candidate tests, confidence intervals, and recursive updates, but provide little explanation or intuition.
There is no worked example that illustrates the algorithm’s reasoning on a small domain, so it is difficult to build an understanding of how rules evolve or how the algorithm achieves its claimed stability.


2. Independence assumption across attributes
The model decomposes transition prediction into independent rules for each object attribute and class for each action.
This implies that attributes vary independently given the action, which is an extremely restrictive assumption for structured environments where attributes are correlated.
No justification, ablation, or discussion is provided for this modeling choice.
The paper should either analyze the implications of this assumption or relax it to model dependencies among attributes.

3. Folding cumulative reward into the state
The paper introduces a special "game" object with an attribute "score" that accumulates the total reward over time.
This means that the state includes historical information, which breaks the Markov property of standard MDPs.
The inclusion of cumulative reward is motivated as a way to avoid learning a separate reward model, but this introduces conceptual and practical problems.
First, it changes the formal definition of the problem and makes direct comparison with other methods unclear.
Second, it conflates transition modeling and reward modeling, since the same structure is now predicting both environment dynamics and reward accumulation.
The paper does not analyze the theoretical consequences of this choice or show that it is harmless in practice.

4. Limited and incomplete related work discussion
The paper’s discussion of prior work is narrow.
It cites QORA and classic inductive logic programming references but does not connect to extensive research on action model learning, probabilistic STRIPS rules, context-specific effects, and relational model-based reinforcement learning. The distinction between "object-oriented" and "relational" is also unclear.
The proposed framework uses relations among objects and existential quantification, which are standard in relational representations, so the terminology difference is superficial.
This omission weakens the paper’s claims of novelty and situates it poorly within the literature (see references below for a starting point)

---

### Official Review · Reviewer_5ypM · 2025-10-28

**Soundness:** 2
**Presentation:** 2
**Contribution:** 2
**Rating:** 4
**Confidence:** 3

**Summary:**

The paper proposes an easily trainable model with interpretability by generating first-order logical decision trees. The performance is evaluated through ablation tests and compared against two baselines: QORA and a neural physics engine. The experimental results indicate that the proposed method, TreeLearn, achieves good performance.

**Strengths:**

The paper conducts several ablation experiments to validate the effectiveness of the proposed method.

**Weaknesses:**

Although the paper states its effectiveness, I believe it lacks adequate discussion of the performance differences among the various baselines. Additionally, some experiments still lack key comparative results. Furthermore, the presentation of this paper still needs to be improved. For example, the paper does not include a discussion of the preliminaries or related work, which would be useful for contextualizing the state-of-the-art models within the inductive logic programming and reinforcement learning research communities.

**Questions:**

1. When analyzing the performance of the neural network baselines, how does the proposed method perform in Figure 5?
2. What is the motivation behind choosing these particular baselines? Are there other baseline models that could be compared?
3. In line 159, what do "deltas" refer to?
4. How are the semantics of nodes determined in the learned first-order decision trees?

---

### Official Review · Reviewer_ae51 · 2025-11-02

**Soundness:** 2
**Presentation:** 2
**Contribution:** 2
**Rating:** 2
**Confidence:** 3

**Summary:**

This paper proposes an algorithm for learning first-order logical decision trees to model transition dynamics in an object-oriented MDP with rewards. The algorithm builds a tree by ~ enumerating logical tests and accepting them if they predict better than the existing tests, and creating new branches out of leaf nodes if this improves accuracy. The algorithm outperforms a prior baseline QORA on a set of dynamics-learning environments. The authors also point out some inference time optimizations you can make for evaluating the learned decision trees.

**Strengths:**

- The approach performs much better than prior work. It can handle variable bindings, nested quantifiers, and more complex rules.
- The algorithm design seems reasonable, although many details of the implementation are not described well.

**Weaknesses:**

Two main weaknesses: impact/significance, and lack of clarity describing the main algorithm.

W1: impact and significance
- Object oriented transition modeling is a domain with limited impact. This is good old fashioned AI, with symbolic observations, learning, and models. It's unclear to me what the insights from 2025 are compared with previous decades of research into this area.
- The evaluation is not very thorough: only one baseline is compared to. The environments are small gridworld environments. Without more evaluation, it's hard to know what the impact of this algorithm will be. Applications? To demonstrate impact, it seems like it could be relevant to compare to research in robotics, which often uses symbolic states and action spaces. However, as presented, it's hard to contextualize the research to the broader field.
- Similarly, it is hard to evaluate the broader strengths of the proposed algorithm. The authors convincingly demonstrate that the algorithm is better than prior work QORA, but that is all.

W2: clarify describing the main algorithm
- the description of the algorithm in the main text is incomplete with regard to many details.
- no pseudocode of the algorithm is provided. Instead, in the appendix, a dump of python implementation is shown, which is highly unusual for a machine learning paper. tedious to read through and obscures the core ideas of the algorithm behind many lines of code. To make the algorithm easier to understand, you should (1) provide pseudocode of the main learning algorithm, (2) defer code to a github link with documentation.
- see questions area for further questions about the algorithm.
- there is no analysis of the complexity of the algorithm

In order for me to accept this paper, I think I would need a convincing argument for the impact of this algorithm, and better contextualization with other symbolic MDP research.

**Questions:**

Q1. How are confidence intervals calculated and used to swap out a rule? The exact mechanism here eluded me.
Q2. When updating nodes, which tests are compared to the current test? I guess you do some enumeration over possible new tests, and evaluate them? Is this described in the main text, or just in the Python code in the appendix?
Q3. On demand queries: "instead of computing every true fact" — is this saying that you're not evaluating all possible enumerable facts? Is there even a finite set of true facts?
Q4. How possible tests and outcomes are enumerated is very unclear. Should be made clearer in the main text.
Q5. How does this algorithm compare to QORA in technical details? Should be made clearer in the main text.

---

### Meta-Review · Area_Chair_TFDb · 2026-01-07

**Summary:**

The paper proposes learning first-order logical decision trees to model object-oriented transition dynamics (and reward) and reports improved performance over prior work such as QORA, with interpretability as a key benefit. Reviewers generally find the idea reasonable and the empirical gains promising. However, the paper is difficult to understand and reproduce due to missing algorithmic details, weak problem framing and related-work positioning, and limited evidence for broad impact.

Pros
1. Interpretable model class based on first-order logical decision trees, which is appealing for structured domains
2. Empirical improvements over the closest baseline (QORA), and some ablations that help validate components
3. Handles richer logical structure than simpler rule learners (e.g., variable bindings, quantifiers)

Cons
1. Method description lacks critical operational details (candidate enumeration, node updates, confidence intervals, on-demand querying); no clear pseudocode, and the appendix code dump is not an adequate substitute
2. Missing analysis of computational complexity and limited intuition or worked examples to clarify how the algorithm behaves
3. Problem setup and positioning are weak: unclear relationship to RL/model-based RL and thin related work given extensive ILP/relational action-model literature
4. Strong modeling assumptions are not justified or ablated (e.g., predicting attributes independently); folding cumulative reward into state raises conceptual issues that are not analyzed
5. Evaluation is narrow (mostly toy/gridworld settings) with limited baselines and insufficient discussion of when/why the method outperforms alternatives, making significance and generality unclear

**Reviewer Concerns:**

No rebuttals

**Reviewer Scores:**

N/A

---

### Decision · Program_Chairs · 2026-01-26

Reject